



# Damage strength increases ice mass loss from Thwaites Glacier, Antarctica

Yanjun Li[1, *], Violaine Coulon[2], Javier Blasco[2], Gang Qiao[3, *], Qinghua Yang[1], and Frank Pattyn[2]

[1]School of Atmospheric Sciences, Sun Yat-sen University, and Southern Marine Science and Engineering Guangdong
Laboratory (Zhuhai), Zhuhai, 519082, China
[2]Laboratoire de Glaciologie, Université libre de Bruxelles, Brussels, 1050, Belgium
[3]College of Surveying and Geo-informatics, Tongji University, Shanghai, 200092, China

*Correspondence to*: Yanjun Li (liyj375@mail.sysu.edu.cn) and Gang Qiao (qiaogang@tongji.edu.cn)

**Abstract.** Ice damage plays a critical role in determining ice-shelf stability, grounding-line retreat, and subsequent sea-level
rise, as it affects the formation and development of crevasses on glaciers. However, few ice-sheet models have explicitly
considered ice damage nor its effect on glacier projections. Here, we incorporate ice damage processes into an ice-sheet model.
By applying the upgraded model to the Thwaites Glacier basin, we further investigate the sensitivity of Thwaites Glacier to
the strength of the ice damage. Our results indicate that the ice-sheet model enabled with the ice damage mechanics better
captures the observed ice geometry and mass balance of the Thwaites Glacier during the historical period (1990–2020),
compared to the default model that ignores ice damage mechanics. Ice damage may result in a collapse of Thwaites Glacier
on multidecadal-to-centennial timescales and a notable increase in ice mass loss. Moreover, ice mass loss from Thwaites
Glacier to the ocean may induce a sea-level rise of 5.0 ± 2.9 cm by 2300, which is more than double the simulation result
without ice damage. This study highlights the importance of explicitly representing ice damage processes in ice-sheet models.

## 1 Introduction

Damage of glaciers is getting more attention due to its impact on glacier and ice sheet evolution with a warming climate.
Previous studies revealed that damage of glaciers could trigger the formation and propagation of rifts and crevasses, which
increases the instability of ice shelves by enhancing shearing in the ice-shelf area, weakening the ice-shelf structure, inducing
additional damage and the retreat of the grounding line (Sun et al., 2017; Lhermitte et al., 2020; Izeboud and Lhermitte, 2023).
Moreover, the ability of ice shelves to restrain the flow of ice from the upstream grounded glaciers towards the ocean (through
the buttressing effect) weakens, leading to an acceleration of grounded ice mass loss and subsequent sea level rise. Damage of
glaciers is a precursor of ice-shelf disintegration and might affect the timing and magnitude of grounded ice loss, as well as
the contribution of Antarctic glaciers to sea-level rise (Lhermitte et al., 2020; van de Wal et al., 2022; Izeboud and Lhermitte,
2023).

Damage of glaciers has only been incorporated into a few ice-sheet models to explore its potential impact on the ice-sheet
dynamics under hypothetical ideal geometry conditions. Sun et al. (2017) coupled a continuum damage mechanics (CDM)



model with an ice-sheet model based on the zero-stress Nye approach (Nye, 1957). By applying the model to an ideal ice sheet geometry (with retrograde bed slopes and strong lateral stress (Gudmundsson et al., 2012)) created by the Marine Ice Sheet Model Intercomparison Project (MISMIP+; Cornford et al., 2020), they found that ice damage results in a larger retreat of the grounding line compared to the simulation without damage. Using the same model, Lhermitte et al. (2020) demonstrated that

intensifying damage at a specific location within the shear zones leads to a broad propagation and amplification of damage throughout the entire shear zone, reinforcing the hypothesis of the positive feedback mechanism. However, the results obtained from tests of the ice-sheet model under ideal geometrical condition might not be fully applicable to the real world, and few studies have investigated the effect of ice damage on the dynamic of real-world ice sheets (e.g., Antarctic glaciers and ice shelves).

Extensive ice damage has been observed on Thwaites Glacier (TG), the largest ice stream in West Antarctica ($2.1 \times 10^5$ km$^2$) and one of the fastest mass-losing outlet glaciers of the Antarctic Ice Sheet (AIS) (Rignot et al., 2019; Lhermitte et al., 2020; Surawy-Stepney et al., 2023a). Recent satellite images show an increase of ice-shelf damage in TG (Bradley et al., 2023), with open rifts and dense crevasses distributing across its floating ice shelf (Thwaites Eastern Ice Shelf (TEIS) and Thwaites Western Glacier Tongue (TWGT)), as well as in the shear zones of both ice shelves (Lhermitte et al., 2020). Episodic dynamic

changes in TWGT, such as acceleration, have been proven to be linked to this damage. Miles et al. (2020) found the rapid acceleration periods identified from 2006 to 2012 and 2016 to 2018 corresponded to structural weakening. Surawy-Stepney et al. (2023a) also confirmed the formation and development of crevasses along the shear margin of the TWGT from June 2017 to December 2018 and in early 2020 consistent with the acceleration of ice flow during these periods. Moreover, as a marine glacier (i.e., grounded below sea level; Fig. 1a) over a retrograde bed slope, TG is susceptible to marine ice-sheet instability

(MISI) (Schoof, 2007; Pattyn 2018). Ice damage may facilitate the grounding line retreat of TG by undermining the structural integrity of ice shelves and reducing their buttressing effect on upstream glaciers. However, Gudmundsson et al. (2023) found that the TEIS is not giving any buttressing to the ice sheet, meaning the loss of this ice shelf would not have a major impact. Despite this, it remains imperative to consider the damage processes when modelling and projecting the evolution of TG under future climate change, as well as the contribution of ice mass loss from the TG basin to global sea level rise.

In this study, the numerical ice-sheet model Kori-ULB (Pattyn, 2017; Coulon et al., 2024), which explicitly represents the continuum damage mechanics (Sun et al., 2017), is employed to investigate the effect of damage on present-day and near-future evolution of the TG basin. We aim to (i) evaluate and calibrate the Kori-ULB model using the observational data on the contribution to sea-level rise and the net mass balance in the TG basin; (ii) quantify the effect of damage on the grounding-line retreat, ice velocity and mass change of the TG basin with a historically calibrated ensemble; and (iii) explore the

sensitivity of the glacier retreat and mass loss in the TG basin to increased damage strength.



## 2 Methods

### 2.1 Ice sheet and damage model

The Kori-ULB ice-sheet model (Pattyn, 2017; Coulon et al., 2024) is a 2.5D thermomechanical finite difference model that combines Shallow Ice Approximation with Shallow Shelf Approximation (so-called hybrid model; Winkelmann et al., 2011). The Kori-ULB model has been proven to be an effective tool for large-scale simulations of the Antarctic Ice Sheet (Seroussi et al., 2020; Coulon et al., 2024). It can also be applied to small drainage basins with divergent ice geometries, such as the hypothetical ice geometries proposed by the MISMIP3d (Pattyn et al., 2013) and MISMIP+ (Cornford et al., 2020) experiments, and the drainage basins in real world (e.g., Thwaites Glacier basin; Kazmierczak et al., 2024).

To investigate the responses of ice dynamics, grounding-line retreat and mass change in the TG basin to ice damage and damage parametric perturbations, we couple the ice-sheet model to the continuum damage model CDM. Damage ($d(\tau_1)$) in CDM includes a local source of damage ($d_l(\tau_1)$) and damage conservation during ice flow ($d_{tr}$). Damage conservation during ice flow ($d_{tr}$) describes the evolution of the vertically integrated damage field caused by advection, stretching, and mass loss or accumulation on the upper and lower surfaces of the glacier, which can be solved by a damage transport equation (Sun et al., 2017). The local source of damage $d_l(\tau_1)$ can be described by the total depth of the crevasses (Nick et al., 2011, 2013; Cook et al., 2014), which includes the depth of surface crevasses $d_s$ and the depth of basal crevasses $d_b$, and can be calculated by the zero-stress rule (Nye, 1957; Nick et al., 2011):

$$d_s = \frac{\tau_1}{\rho_i g} + \frac{\rho_w}{\rho_i} d_w, \tag{1}$$

$$d_b = \frac{\rho_i}{\rho_w - \rho_i} \left( \frac{\tau_1}{\rho_i g} - H_{ab} \right), \tag{2}$$

$$d_1(\tau_1) = max\left(0, \left( (d_s, d_s + d_b), C_1 * h \right)\right), \tag{3}$$

where, $d_w$ is the water depth in the surface crevasse (here we only consider dry crevasses, so $d_w$ is equal to 0), $H_{ab}$ is the thickness above floatation, $g = 9.81$ m s$^{-2}$ is the gravitational acceleration, $\rho_i = 917$ kg m$^{-3}$ and $\rho_w = 1028$ kg m$^{-3}$ are the ice and seawater density, respectively. $\tau_1$ is the first principal stress, $h$ is ice thickness and $C_1$ is a damage parameter that describes the upper limit of $d_l(\tau_1)$ as a fraction of the ice thickness. The final relationship of damage ($d(\tau_1)$) is expressed as:

$$d(\tau_1) = min(C_{tr} * h, max(d_1(\tau_1), d_{tr})), \tag{4}$$

where, $C_{tr}$ is a second damage parameter that describes the upper limit of $d(\tau_1)$ as a fraction of the ice thickness. $C_1$ is equal to or less than $C_{tr}$. A comprehensive description of the Kori-ULB ice-sheet model and its integration with the CDM model is given in Appendix A.



## 2.2 Model initialization and simulation protocol

The Kori-ULB model uses the present-day ice sheet surface and bed geometry and grounding-line location from Bedmachine
90   v2 (Morlighem et al., 2020) as input. Ice sheet initial conditions are obtained through the equilibrium initialization strategy by
an inverse simulation nudging towards present-day ice-sheet geometry (Pollard and DeConto, 2012; Bernales et al., 2017;
Coulon et al., 2024). This results in an undamaged present-day steady state for 1990 (as shown in Fig. A1). After model
initialization, the root mean square errors (RMSEs) between simulated and observed ice velocity and ice thickness are 201 m
a$^{-1}$ (786 m a$^{-1}$ for the floating ice) and 28 m (43 m for the floating ice), respectively. The grounding-line position of the TG
95   basin closely matches the observed grounding-line position (Gardner et al., 2018).

Starting from this initial state, we conduct an ensemble of simulations to explore the impact of ice damage on the dynamic
evolution of the TG basin. We design a perturbed parameter ensemble including the two key parameters $C_l$ and $C_{tr}$ (Eqs. 3 and
4) that govern the damage feedback processes. $C_l$ sets a limit on local damage and $C_{tr}$ sets a limit on total damage. We initially
create a 100-member ensemble by sampling $C_1$ and $C_{tr}$ within the range of 0–1 using a Latin hypercube sampling method. The
100   members in our ensemble are subsequently reduced to 43 to meet the requirement that $C_1 < C_{tr}$. Finally, the ensemble with 43
parameter members is used to quantify the sensitivity of TG evolution to the strength of ice damage.

We conduct an ensemble of historical simulations for 30 years (1990–2020) under present-day conditions with each of the
43 parameter members (Table A1). Based on the results from these historical simulations, the parameter values of $C_l$ and $C_{tr}$
are further constrained by the satellite-based estimate of ice mass change in the TG basin (Shepherd et al., 2019). Parameter
105   members that enable the simulated ice mass change (the contribution to sea-level rise and net mass balance) in the TG basin
being within the range between the mean value of satellite-based estimate ± two times of the observed standard deviation (s.d.)
are considered appropriate. These parameter members are then classified as Group 1 (G1). If the tested parameter member
drastically over- or underestimates (beyond +/- 2 s.d.)) the observed ice mass change in the TG basin, it is classified as Group
2 (G2).

Two control simulations (Table 1 and Table A1) without damage are used as a baseline for comparison. One is designed to
reproduce the observed mass change rates (the Ctrl$_{cal}$ experiment), while the other is not (the Ctrl experiment). In the Ctrl$_{cal}$
experiment, we force the model to reproduce the historical trends by integrating satellite-based data of present-day ice mass
change rates in the TG basin (Otosaka et al., 2023), thereby facilitating ice thinning at the beginning of the run. The Ctrl$_{cal}$
experiment matches observed trends relying on different physics compared to the damage experiments. Consequently, the
differences in the dynamic changes of TG between experiments with or without considering damage can be used to quantify
the impact of damage strength on its future dynamics.

We employ the method described in van den Akker et al. (2024) to derive the initial state of the Ctrl$_{cal}$ experiment (Fig. A2).
After model initialization, RMSEs of ice velocity and thickness for this initial state are 172 m a$^{-1}$ (659 m a$^{-1}$ for the floating
ice) and 27 m (54 m for the floating ice), respectively. The initial grounding-line position also closely matches the observed
grounding-line position. At the start of the historical run, the present-day SMB is reinstated without the additional mass-change



term. Hence, by construction, the simulated ice sheet reproduces the observed mass-change rates. Note that the initial state for the Ctrl experiment is the same as that used in the damage sensitivity experiments. After the historical period, simulations are extended until the year 2300, with constant atmospheric and oceanic forcing at present-day conditions, to assess the effect of damage and the potential response and sensitivity of TG evolution to the strength of ice damage at larger time scales.

**2.3 Forcing data**

Initial present-day surface mass balance and temperature are obtained from the polar regional climate model MARv3.11 (Kittel et al., 2021). Present-day ocean temperature and salinity are derived from data provided by Schmidtko et al. (2014). Please see Table A2 for all forcing and model calibration and evaluation data used in this study. The basal melting underneath the floating ice shelves is estimated using the PICO model by Reese et al. (2018). All simulations in this study are performed at a spatial

resolution of 2 km.

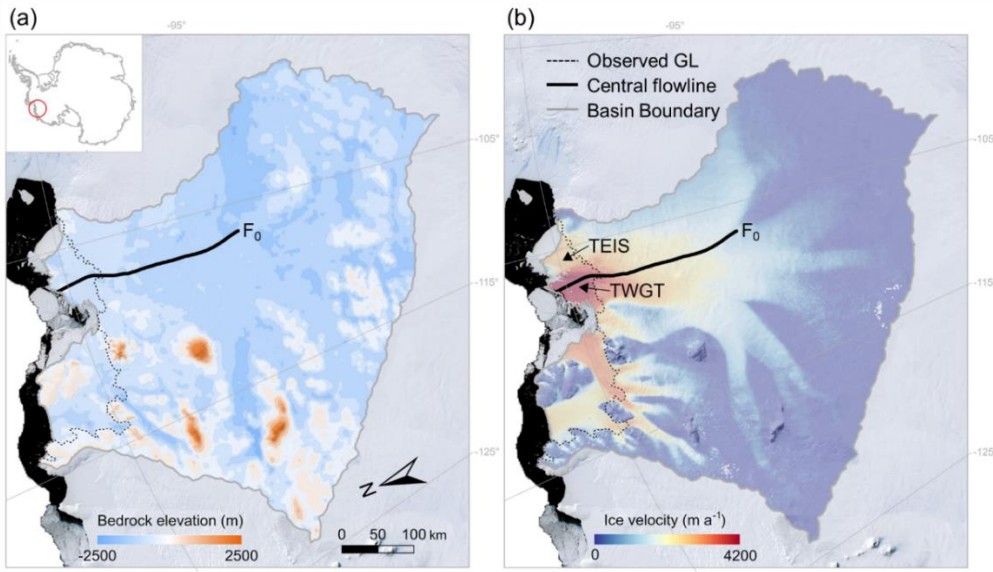

**Figure 1.** Bedrock elevation and ice velocity in the TG basin. (a) Observed bedrock elevation of the TG basin based on BedMachine v2 data (Morlighem et al., 2020) and (b) observed ice velocity of the TG basin based on MEaSUREs data (Rignot et al., 2017) overlapped on the Landsat image mosaic of Antarctica (LIMA) mosaic (Bindschadler et al., 2008). The black solid curve is the central flowline profile
stemming from the Antarctic surface flowline dataset developed by Liu et al. (2015), which spans 340 km from the inland grounded ice ($F_0$) to the calving front. The black dashed line shows the position of the observed grounding line (Gardner et al., 2018). The inset shows the location of the TG basin in Antarctica. TEIS represents the Thwaites Eastern Ice Shelf and TWGT represents the Thwaites Western Glacier Tongue.



## 3 Results

### 3.1 Effects of ice damage on the simulated historical evolution of Thwaites Glacier

The 45-member ensemble of simulations over 1990–2020 in the TG basin show a high sensitivity of mass loss to the strength of ice damage (Fig. 2). The simulated contribution of ice mass change ranges from -291.54 to -9.37 Gt a$^{-1}$. Of all 43 parameter members of $C_1$ and $C_{tr}$, 15 members are classified into Group 1 (Table 1 and Table A1, light green lines in Fig. 2). The remaining 28 members are classified into Group 2 (light red lines in Fig. 2). In Group 1, the simulated net ice mass change in
the TG basin ranges from -54.28 to -31.75 Gt a$^{-1}$, with the highest mass loss estimate being 1.7 times of the lowest estimate.

**Table 1.** Summary of the typical damage sensitivity experiments and two control experiments performed at the TG basin.

| Experiments | Description | Damage parameters | |
| --- | --- | --- | --- |
| | | $C_1$ | $C_{tr}$ |
| Ctrl | deactivated damage processes | - | - |
| Ctrl$_{cal}$ | deactivated damage processes & satellite-observed mass balance calibrated (Otosaka et al., 2023) | - | - |
| Group 1 | damage processes & SLC and net mass balance within the range of observational estimates ± 2 s.d. (0.24 ± 0.08 cm and -46.1 ± 14.4 Gt a$^{-1}$ over 1992–2017) in the historical simulation (Shepherd et al., 2019) | [0–0.24] | [0.2–0.8] |
| Group 2 | damage processes & SLC and net mass balance outside the range of observational estimates ± 2 s.d. in the historical simulation | [0–0.56] | [0–1] |

The explicit representation of ice damage processes better captures the observed ice mass change in the TG basin compared to the default model without damage (Ctrl experiment in Fig. 2). The simulated mean value of mass change for Group 1 is -38.3 Gt a$^{-1}$, which is comparable to satellite-derived observations (-46.1 ± 14.4 Gt a$^{-1}$; mean ± 2 s.d.). Ignoring ice damage
underestimates ice mass change (-2.1 Gt a$^{-1}$; Ctrl experiment in Fig. 2). In contrast, the Ctrl$_{cal}$ experiment, which uses an artificial calibration of the ice mass change rate, reproduces a simulated mass change (-28.1 Gt a$^{-1}$) comparable to the estimates for Group 1.

Ice damage processes also result in a larger grounding-line retreat (Fig. 2c and Fig. A3). By 2020, the simulated grounding lines by the ensemble of Group 1 (the green lines in Fig. 2c) retreat by 6-10 km further along the central profile than the
observed grounding-line position (Gardner et al., 2018; the black dashed line in Fig. 2c). All simulated grounding lines in Group 2 also show a larger retreat by 2020 than the observed grounding-line position (Fig. 2c), with the maximum retreat being about 44 km along the central profile and a retreat rate of up to 1.5 km a$^{-1}$ (the red line in Fig. A3). This retreat rate is twice the observed mean retreat rate (~0.7 km a$^{-1}$) over 1992–2011 (Rignot et al., 2014) and 2 to 5 times of the observed annual retreat rate (0.3–0.6 km a$^{-1}$) over 2011–2017 (Milillo et al., 2019). In addition, the grounding-line positions simulated by the
two control experiments that ignore damage show an overall retreat during the historical simulation period (Fig. A3). The simulated grounding-line position in the Ctrl$_{cal}$ experiment shows a retreat comparable to those simulated in the Group 1



experiments in the eastern section of TG. In contrast, the retreat simulated in the Ctrl experiment is relatively minor. Along the central profile of TG, the simulated grounding-line positions of the two control experiments are even more seaward than the satellite-based observation (Fig. 2c and Fig. A3).

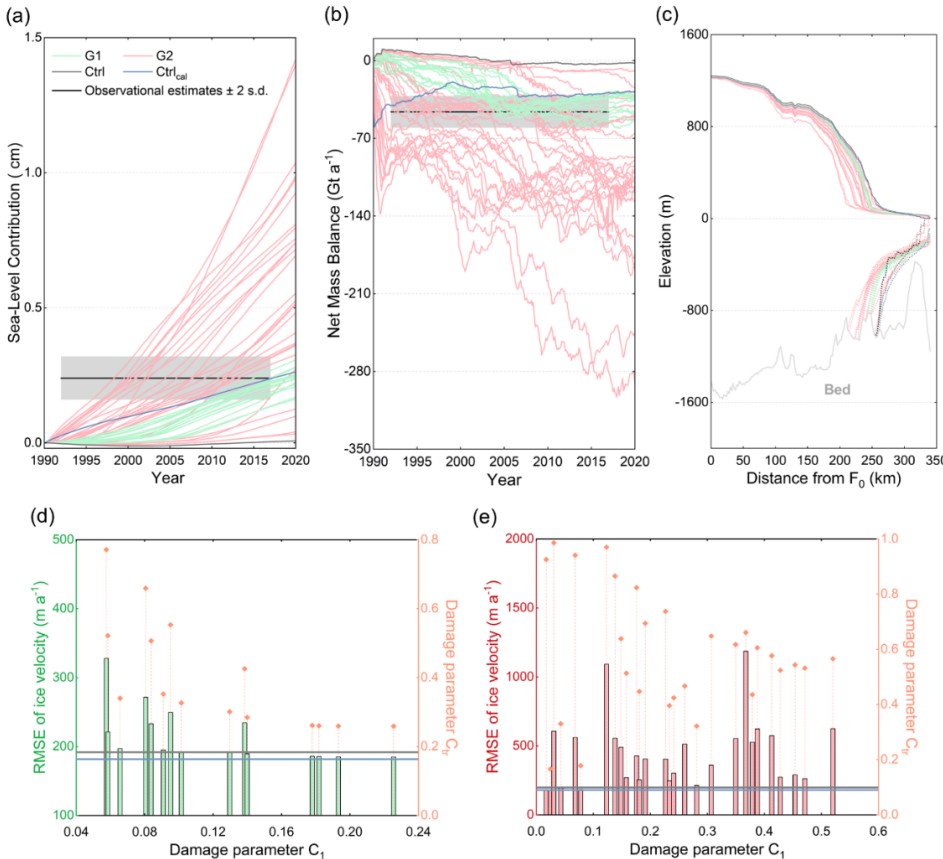


**Figure 2.** The simulated change trends of ice mass balance and grounding-line position in the TG basin under different damage strengths over the period 1990–2020. (a) Contribution of ice mass loss in the TG basin to global sea-level rise; (b) the net mass balance (considers volume above flotation only, i.e., the rate of mass change contributing to sea-level rise) in the TG basin; (c) the geometry profiles along the central flowline profile (the black curve in Fig.1) and the simulated (red and green dashed lines) and observed (black dashed line) grounding-
line positions. RMSEs between the simulated and observed ice velocity under different parameter combinations of $C_1$ and $C_{tr}$ in (d) Group 1 and (e) Group 2. The black lines and shaded areas in (a) and (b) represent the observed mean value ± 2 standard deviation (Shepherd et al., 2019). The grey line represents the simulation result by the model that ignored ice damage processes and did not integrate satellite-based observation of present-day mass change rates to constrain the model initialization (Ctrl experiment), and the blue line represents the simulation result by the model that ignored ice damage processes but integrated satellite-based observation to constrain the model
initialization (Ctrl_cal experiment).

The RMSEs between the observed and simulated ice velocities in the Ctrl_cal and Ctrl experiments are 181 m a$^{-1}$ (753 m a$^{-1}$ for the floating ice) and 191 m a$^{-1}$ (745 m a$^{-1}$ for the floating ice), respectively (Fig. 3). Incorporation of ice damage induces a notable change in the simulated ice flow velocity over the historical period (Figs. 2d–2e and Fig. 3). By 2020, the mean RMSEs of simulated ice velocities in Group 1 and Group 2 simulations are 216 ± 40 m a$^{-1}$ (897 ± 177 m a$^{-1}$ for the floating ice) and
441 ± 245 m a$^{-1}$ (1693 ± 877 m a$^{-1}$ for the floating ice), respectively. This suggests that the parameter members, which enable




to reasonably capture the observed ice mass balance in the TG basin (i.e. the ensemble of Group 1), also ensure that the model can better reproduce the observed ice velocity. For all Group 1 simulations with activated damage processes in the model, the RMSE between observed and simulated ice velocity is the lowest when $C_l$ and $C_{tr}$ are set to 0.23 and 0.26, respectively.

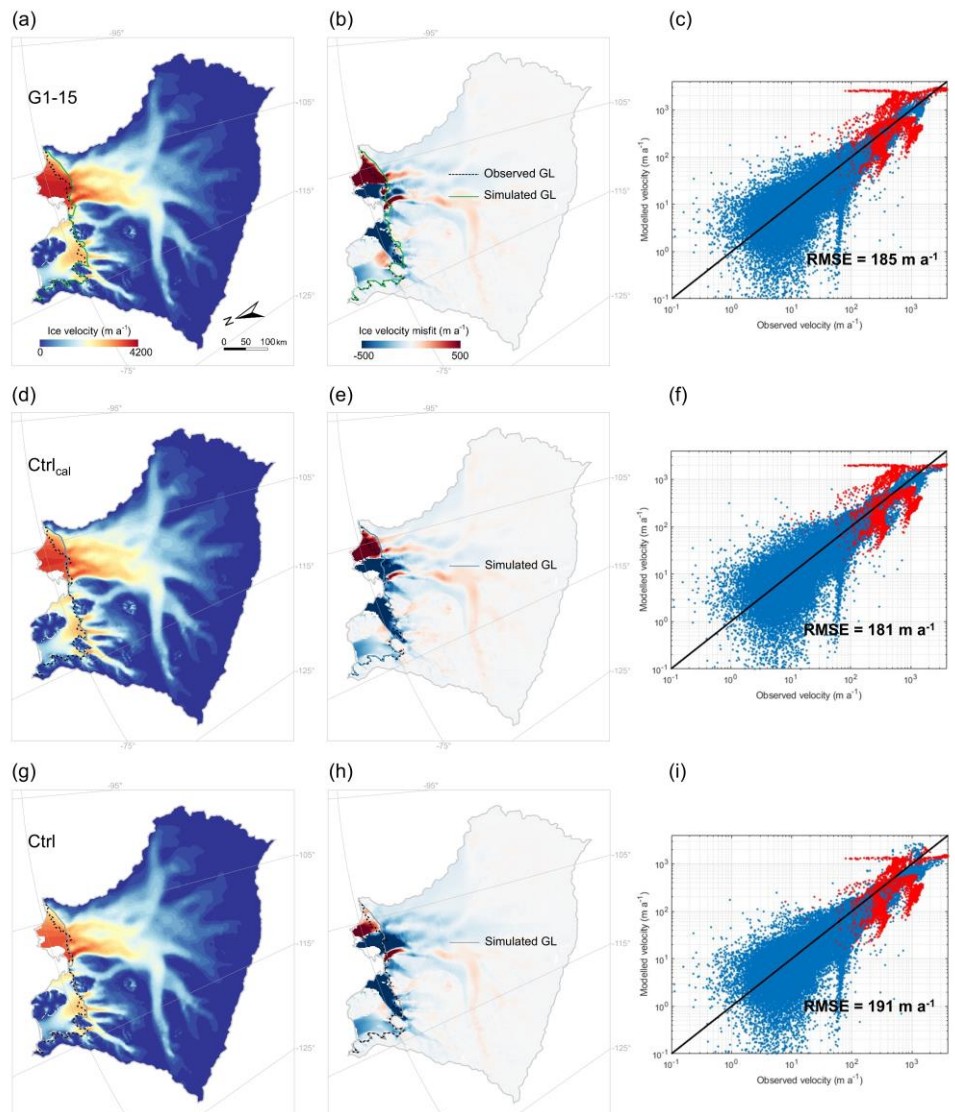

**Figure 3.** The simulated ice velocity under different simulation experiments over the historical period 1990–2020. G1-15 denotes the simulation experiment in the ensemble of Group 1 ($C_l$ =0.23, $C_{tr}$ =0.26) which gives the most accurate (lowest RMSE) simulation result of ice velocity. The Ctrl$_{cal}$ and Ctrl are the two simulation experiments by the model with deactivated damage processes (see Methods for details). (a), (d) and (g) show the spatial distribution of simulated ice velocity in the TG basin by different simulation experiments. (b), (e) and (h) show the difference between simulated and observed ice velocities. (c), (f) and (i) show the comparison between simulated and observed ice velocities at each grid cell in the TG basin, with blue and red dots representing the grid cells of grounded ice and floating ice, respectively. In all maps, the black dashed line is the observed grounding line (Gardner et al., 2018), and the solid lines are simulated grounding lines.




### 3.2 Effect of ice damage on the future evolution of Thwaites Glacier

Comparison of projection results over the period 2020–2300 indicates that ice damage leads to an increased ice velocity,
reduced ice thickness, an accelerated retreat of the grounding line, as well as an increased ice mass loss (Fig. 4). The simulated
mean ice velocity along the central profile of the TG basin (the black line in Fig. 1) by simulations in Group 1 increases from
$236 \pm 113$ m a$^{-1}$ for the grounded ice sheet to $3368 \pm 936$ m a$^{-1}$ at ice front, which is 2–3 times of the estimates from control
simulations (Fig. 4c). The simulated mean ice thickness along the central profile by simulations in Group 1 (from 2506 m at
inland to 108 m at ice front) is approximately 200 m thinner than the result of the control simulations (Fig. 4d).

With ice damage, the simulated grounding lines of TG retreat further inland than the simulation results without ice damage
processes (Figs. 4a and 4b). Moreover, the simulated grounding line retreats to a retrograde-slope bed along the central profile
when damage is accounted for (Fig. 4b), suggesting further inland retreat influenced by ice sheet collapse when the ice shelf
becomes weak enough, indicating instability in the TG basin. In contrast, the simulated grounding lines of the control
experiments are positioned at a pinning point, which is less susceptible to sustained grounding-line retreat and thus enhances
their stability.

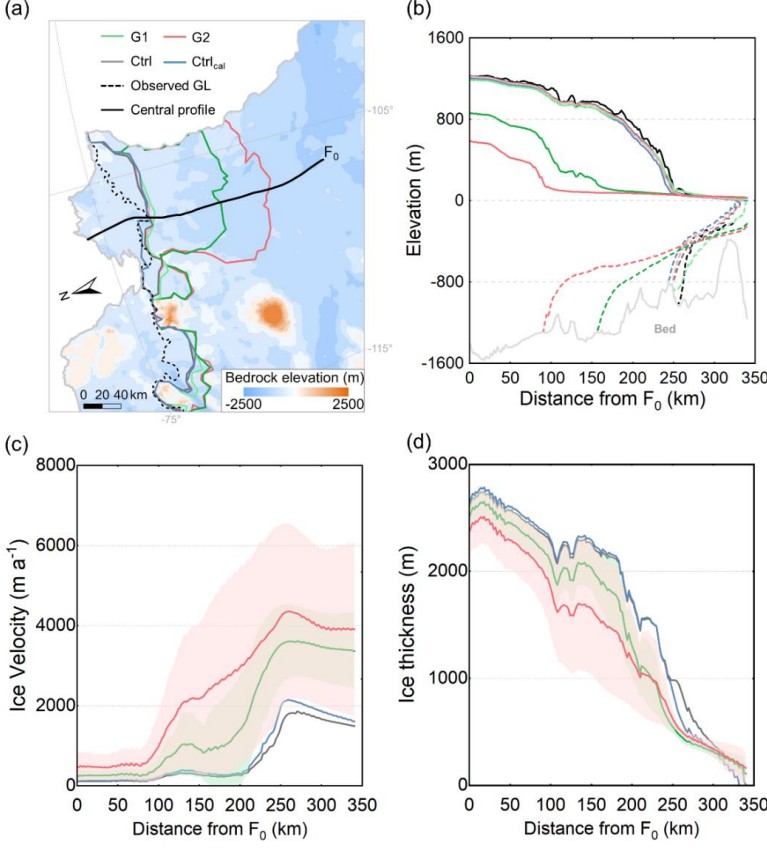

**Figure 4.** (a) Spatial evolution of grounding-line position. Evolution of (b) the ice geometry, (c) ice velocity, and (d) ice thickness along the
central profile (the black curve in (a)) in TG by 2300. The light green (red) and dark green (red) lines in (a) and (b) represent the experiments





with the least and the most retreats of the grounding line in Group 1 (Group 2), respectively, which are also corresponding to the experiments
with the lowest and highest damage strength in Group 1 (Group 2). The black dashed line presents the observed grounding-line position (Gardner et al., 2018). The background figure in (a) is the observed bedrock elevation of the TG basin derived from BedMachine v2 data (Morlighem et al., 2020). In (c) and (d), the solid line represents the mean and the hatched area represents the ensemble standard deviation. The blue and grey lines present simulated results (e.g., the simulated grounding-line position, ice velocity and ice thickness) of the $\text{Ctrl}_{\text{cal}}$ and Ctrl experiments, respectively.

By 2300, the simulated mean net mass loss from simulations in Group 1 reaches $-110 \pm 65$ Gt a$^{-1}$, which is 6–9 times of the

estimates from control simulations (Fig. 5c). The simulated mean decrease of grounded ice area is $7243 \pm 2874$ km$^2$ compared

to 2420–2904 km$^2$ for the simulation without ice damage (Fig. 5b). Corresponding to the increase of ice mass loss, damage

processes tend to induce a larger contribution of the TG basin to global sea level rise (Fig. 5a). In year 2300, the simulated

mean contribution of ice mass loss in the TG basin to global sea level rise by simulations in Group 1 is $5.0 \pm 2.9$ cm, higher

than the simulation results from the Ctrl (1 cm) and the $\text{Ctrl}_{\text{cal}}$ (2 cm) experiments (Fig. 5a).

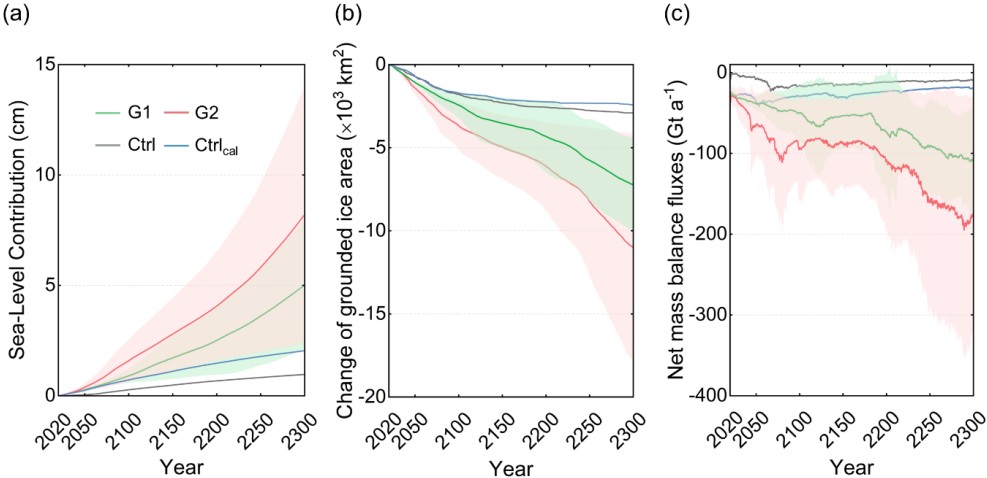

**Figure 5.** Evolution of (a) the contribution to global mean sea-level rise, (b) the change of grounded ice area, and (c) the net mass balance (considering volume above flotation only, i.e. the rate of mass change contributing to sea-level rise) of the TG basin over the projection period 2020–2300 under constant present-day conditions. Solid line represents mean, hatched area represents ensemble standard deviation.

Our simulation results show the increase of damage fraction from the grounded glacier to the front of the ice shelf (Fig. 6).

Near the grounding line, the damage fraction (vertically averaged damage, $D = d/h$) remains relatively low (0.1 in lower

damage strength to 0.4 in higher damage strength). This could be attributed to the combined effects of low viscous stress and

ice overburden counteracting basal crevasse formation (Sun et al., 2017). As damage is transported with the ice flow, this

fraction increases towards the ice front (0.3 in lower damage strength to 0.7 in higher damage strength), with a pronounced

increase in the shear zone where high damage strengths concentrate.

Moreover, our results reveal strong positive feedback between the damage processes and ice-shelf weakening in the TG

basin (Fig. 6). Damage induces ice-shelf weakening and acceleration in the TG basin, which subsequently leads to ice thinning

and the grounding line retreat. The increased ice velocity and decreased ice thickness further stimulate damage formation and

propagation. For instance, in the simulation experiment with the highest damage strength in Group 2 (right panel in Fig. 6),



ice thickness declined by up to 450 m along the grounding line (Fig. 2c) and grounding-line retreats by approximately 16 km during 1990–2020 (Fig. 6c). By 2300, ice thickness declined by ~1300 m around the grounding line (Fig. 6i), and the grounding-line retreats by 148 km, along with the propagation of the damage area (Fig. 6f). A recent finding indicates that Thwaites Glacier exerts a limited buttressing effect on the upstream grounded ice (Gudmundsson et al., 2023). Our results suggest that although damage formation was confined to the floating ice shelf, the observed thinning of the upstream grounded

ice sheet implies that damage on the ice shelf already impacts the upstream grounded ice, which has the potential to induce a remarkable retreat of the grounding line in the future.

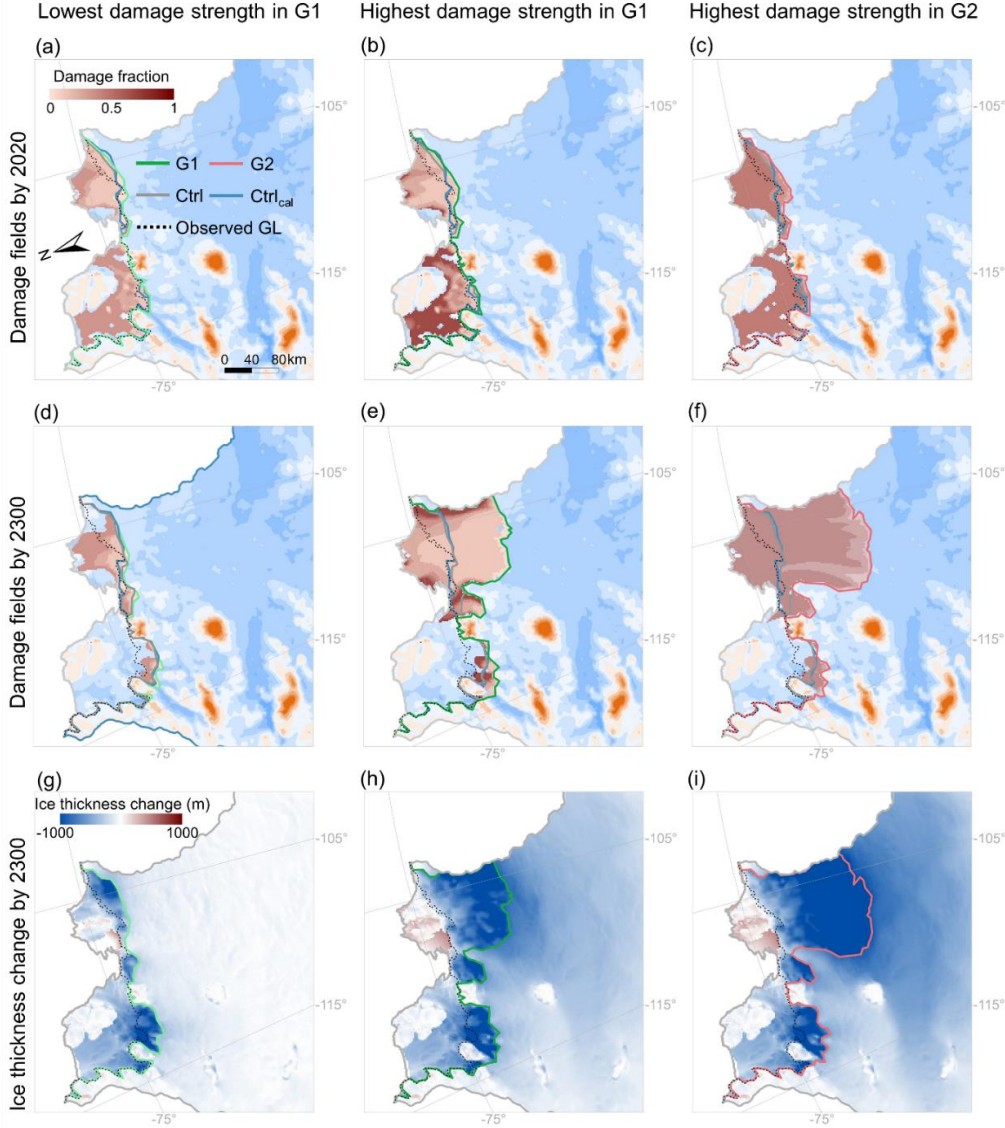

**Figure 6.** Damage fields in the year 2020 (the upper panel) and 2300 (the middle panel) under varying damage strengths, and the resulted ice thickness change (the lower panel). Maps in the first, second and third columns show simulation results under the lowest damage strength





in Group 1 (G1), the highest damage strength in G1 and the highest damage strength in Group 2 (G2), respectively. The black dashed line presents the observed grounding-line position (Gardner et al., 2018). The blue and grey lines present simulated grounding-line positions by the Ctrl$_{cal}$ and Ctrl experiments, respectively.

## 4 Discussion

Previous applications of damage models on an idealized geometry created by the MISMIP+ project showed that the grounding
line retreated significantly, while calving has a smaller influence on the grounding-line retreat than basal melt (Sun et al., 2017, Lhermitte et al., 2020). The strong back stress exerted by the sidewall of stat idealized basin may limit calving and the resulting ice mass loss (Sun et al., 2017). However, our simulations applied to the TG basin reveal a substantial ice mass loss due to calving, particularly under higher damage strength (Fig. A4). This finding highlights the necessity to investigate the effect of damage on ice shelves for real-world cases.

Consistent with recent observations (Rignot et al., 2019), our simulation results suggest that the ice mass loss in the TG basin is primarily driven by sub-shelf melt (Fig. A4). Sub-shelf melt thins the ice shelf, which subsequently weakens the buttressing effect of the ice shelf to the upstream grounded ice sheets, accelerating the ice flow from the upstream glacier to the ocean (Gudmundsson et al., 2019). With increasing damage, the ice mass loss caused by calving becomes a key contributor to the total ice mass loss in the TG basin (Fig. A4). By the year 2300, the mean simulated ice mass losses in the Group 1
experiments due to sub-shelf melt and calving are 129 and 106 Gt a$^{-1}$, respectively. The sum of ice mass loss caused by sub-shelf melting and calving far exceeds the ice mass accumulation on the surface of the TG, resulting in a net mass loss from the TG basin.

By arbitrarily calibrating the initial state of the TG basin using the observed mass change rates (i.e., the Ctrl$_{cal}$ experiment), the non-damage model captures the observed ice geometry, mass balance and ice velocity in the year 2020 rather well, similar
to the model with damage (Fig. 2). However, the long-term projections of TG's evolution over 2020-2300 in the Ctrl$_{cal}$ experiment differ significantly from the results simulated by the model that explicitly represents the ice damage processes. This reveals that having an accurate initial state alone is insufficient, and it is necessary to comprehensively incorporate as many key processes that affect the dynamics of glaciers and ice shelves as possible, including the ice damage, into ice-sheet models.

In addition, although the simulated historical state in the TG basin is overall consistent with observations when damage strength is properly represented in our model, there are still potential uncertainties in the projected evolution of the TG basin over 2020–2300. Firstly, we adopted the mean boundary conditions over 1995–2014 (i.e., the present-day climate condition including the surface mass balance and temperature obtained from the polar regional climate model MAR (Kittel et al., 2021) and present-day ocean temperature and salinity on the continental shelf derived from Schmidtko et al. (2014)) to initialize the
model and simulate the historical and future evolution of the TG basin. These boundary conditions do not necessarily represent the real imbalance of the ice sheet for that period. Secondly, we did not account for damage healing, which may result in an overestimation of the damage field (Sun et al., 2017). Previous studies have found that crevasses can be healed, in response to



overburden pressure, surface ice accumulation, and refreezing (Albrecht and Levermann, 2012; Surawy-Stepney et al., 2023b). Dense crevasses near the grounding zone can be healed during their advection towards the calving front. The healing process
of crevasses can occur when shearing stress along the flow path decreases notably (Wesche et al., 2013; Benn and Åström, 2018). However, studies on the process of ice healing are still scarce due to the challenges in monitoring and quantifying this process (Albrecht and Levermann, 2012).

This study focuses specifically on damage and its influence on the ice sheet stability, but ignores the potential effect of hydrofracturing and marine ice-cliff instability (MICI). Previous studies have shown that hydrofracturing resulting from
surface melting plays a vital role in ice-shelf disintegration (DeConto and Pollard, 2016; Laffin et al., 2022, Bassis and Walker, 2012, Bassis et al., 2021). Pollard et al. (2015) found that hydrofracturing and MICI drastically accelerate the collapse of the West Antarctic Ice Sheet in several decades (Pollard et al., 2015). Similar to Sun et al. (2017), the CDM used in this study only considers dry crevasses, hence ignores hydrofracturing. This may result in an underestimation of ice velocity and ice mass loss from the TG basin in our simulation results. However, recent studies suggest that Thwaites Glacier might be less vulnerable
to MICI than previously thought, and the intrusion of warm seawater or ice sheet surface melt could substantially enhance the response of marine ice sheets to climate change by increasing melting and slipperiness (Morlighem et al., 2024; Robel, 2024). This increased melting can lead to substantial ice damage. Our results also indicate that ice damage could be an alternative process to explain the rapid ice loss of Thwaites Glacier. Furthermore, the lack of representation of some other processes, such as basal hydrological processes, the accretion of marine ice within basal crevasses (Sun et al., 2017) and plastic necking (Bassis
and Ma, 2015), is also potential to induce some uncertainties in our simulation results.

Our study exclusively investigates the sensitivity of ice dynamics (e.g., the grounding-line retreat, ice velocity and ice thickness) and ice mass change in the TG basin to the damage strength. Nevertheless, these findings may not hold true in other basins in Antarctica. Thus, it is necessary to apply our model to more basins with different climatic, geometrical and oceanic conditions. Moreover, it is also important to investigate the influence of damage on the evolution of the Antarctic ice sheet
under different climate change scenarios (Seroussi et al., 2020). Such a comprehensive study is vital for accurately predicting the future evolution of the Antarctic ice sheet and its contribution to global sea-level rise under climate change.

## 5 Conclusion

In this study, we performed a comprehensive analysis on the response of Thwaites Glacier to ice damage at different strengths using the Kori-ULB ice-sheet model. Comparison of simulation results from the model with activated and deactivated ice
damage processes indicates that an explicit representation of ice damage in the ice-sheet model allows to better simulate the observed ice geometry and mass balance in the Thwaites Glacier basin. Even starting from a present-day state calibrated against observed ice mass change rates, the projection of the Thwaites Glacier basin's evolution from 2020 to 2300 without ice damage differs significantly from simulations that explicitly represent the ice damage processes. Increased damage strength generally results in larger retreat of the grounding line, higher ice velocity, thinner ice shelves, more ice mass loss, and bigger



contribution to global sea-level rise. This study highlights the necessity for further research on damage processes and the importance of integrating damage into ice-sheet models to more accurately project the future evolution of the Antarctic ice sheet under climate change.

*Code and data availability.* The code and reference manual of the Kori-ULB ice-sheet model are publicly available on GitHub via https://github.com/FrankPat/Kori-dev (last access: 9 September 2024). The specific Kori-ULB model version used in this
study, the simulation outputs will be made available on Zenodo once published. All datasets used in this study are freely accessible through their original references. The MAR outputs used in this study are available on Zenodo (https://doi.org/10.5281/zenodo.4459259; Kittel et al., 2021).

*Author contributions.* YL conceived the study in collaboration with VC, JB, GQ, QY and FP. YL and VC developed the experimental setup and design, with contributions from JB and FP. YL set up the ice-sheet model and performed all model
simulations. YL performed the data analysis, produced the figures, and wrote the original manuscript draft. All authors contributed to designing the simulations and provided feedback on the analysis and input to the manuscript.

*Competing interests.* The authors declare that they have no conflict of interest.

*Acknowledgements.* YL and QY received support from the National Natural Science Fund of China (No. 42406242), the Southern Marine Science and Engineering Guangdong Laboratory (Zhuhai) (Nos. SL2021SP201 and SML2022SP401), and
the Fundamental Research Funds for the Central Universities, Sun Yat-sen University (No.74110-31610046). This research was also supported by OCEAN:ICE, which is co-funded by the European Union, Horizon Europe Funding Programme for research and innovation under grant agreement Nr. 101060452 and by UK Research and Innovation. O:I Contribution number XX. Computational resources have been provided by the Consortium des Équipements de Calcul Intensif (CÉCI), funded by the Fonds de la Recherche Scientifique de Belgique (F.R.S.-FNRS) under Grant No. 2.5020.11 and by the Walloon Region.
JB received support from the HiRISE (NWP GROOT, Netherlands) under grant agreement No. XX and the Nederlandse Organisatie voor Wetenschappelijk Onderzoek (NWO) under grant No. OCENW.GROOT.2019.091.

**Appendix A: Integration of the CDM model into the Kori-ULB ice-sheet numerical model**

We implement the CDM in the Kori-ULB ice sheet numerical model. In Kori-ULB, the relationship between the deviatoric stress $\tau$ and the strain rate $\dot{\varepsilon}$ is described by Glen's constitutive flow law:

$$2A\tau^2\tau = \dot{\varepsilon}\,,\tag{A1}$$

$A$ is Glen's flow law factor. Following Sun et al., (2017) and Bassis and Ma (2015) the propagation of damage reduces the ice viscosity, through Glen's flow law, leading to faster ice flow. Here, a damage factor $D(\tau)$ is introduced in Eq. (A1) to describe this damage feedback process in Kori-ULB:

$$2A\tau^2\tau = (1 - D(\tau))^3\,,\tag{A2}$$





Given the shallow shelf approximation, this results in the following expression for the vertically integrated effective viscosity:

$$2h\mu = (h - \tau_1)A^{-\frac{1}{3}}\dot{\varepsilon}^{-\frac{2}{3}},$$    (A3)

where, $\mu$ is effective viscosity and $h$ is ice thickness. In Kori-ULB, the first principal stress $\tau_1$ and ice velocity $v = (u, v)$ together can be numerically solved using the stress balance equation. In this way, the relationship between the damage

and the first principal stress $\tau_1$ needs to be defined to realize the coupling of the damage with the Kori-ULB ice sheet model. Here, we use CDM to link the damage and $\tau_1$. CDM considers both the local source of damage and its transport during ice flow (Sun et al., 2017). The local source of damage can be described by the total depth of the crevasses (Nick et al., 2011, 2013; Cook et al., 2014), which includes the depth of surface crevasses $d_s$ and the depth of basal crevasses $d_b$, and can be calculated by the zero-stress rule (Nye, 1957; Nick et al., 2011):

$$d_s = \frac{\tau_1}{\rho_i g} + \frac{\rho_w}{\rho_i} d_w,$$    (A4)

$$d_b = \frac{\rho_i}{\rho_w - \rho_i}\left(\frac{\tau_1}{\rho_i g} - H_{ab}\right),$$    (A5)

where, $d_w$ is the water depth in the surface crevasse (here we only consider dry crevasses, so $d_w = 0$), $H_{ab}$ is the thickness above floatation, $g = 9.81$ m s$^{-2}$ is the gravitational acceleration, $\rho_i = 917$ kg m$^{-3}$ and $\rho_w = 1028$ kg m$^{-3}$ are the ice and seawater density, respectively. $\tau_1$ can be calculated by the principal strain $\varepsilon$:

$$\tau_1 = \frac{1}{2}\varepsilon\mu,$$    (A6)

Then, the total local crevasse depth, namely the local damage $d_1(\tau_1)$ can be defined as:

$$d_1(\tau_1) = max\left(0, \left((d_s, d_s + d_b), C_1 * h\right)\right),$$    (A7)

Here, we use damage parameter $C_1$ to describe the upper limit of the local damage $d_1(\tau_1)$ as a fraction of the ice thickness, with the parameter ranging from 0 to 1. If there is no advection, $\tau_1$ can be determined by setting it equal to the overall depth

of crevasses.

The damage transport during ice flow describes the evolution of the damage field due to advection, stretching, and the loss and accumulation of mass on the upper and lower surfaces of the glacier. For any time and position (x, y, t), there is a local damage field $d_1$(x, y, t) and a transport damage field $d_{tr}$(x, y, t), the latter describes the total depth of crevasses after the ice flow process by solving the damage transport equation (Sun et al., 2017):

$$\frac{\partial d_{tr}}{\partial t} + \nabla \cdot (u d_{tr}) = -[(\dot{a}, 0) + (\dot{m}, 0)]\frac{d_{tr}}{h},$$    (A8)

The left-hand side of Eq. (A8) represents the vertically integrated damage conservation under ice flow, which includes the movement of the crevasses along with the ice flow and the stretching and compression. On the right-hand side, an increase in



undamaged ice thickness is presumed to occur due to accumulation on the upper surface ($\dot{a}$), while the crevassed underside is eroded by basal melting ($\dot{m}$). Regardless of whether the horizontal flow field is divergent or convergent, all these factors

maintain a constant ratio of $d_{tr}$ to $h$ (Sun et al., 2017).

Assuming that at least during the timescale of the closure process, the crevasse surfaces do not bond together as a result of crevasse closure, the final relationship $d(\tau_1)$ is expressed as:

$$d(\tau_1) = min(C_{tr} * h, max\,(d_1(\tau_1), d_{tr}))\ , \tag{A9}$$

By bringing Eq. (A7) into Eq. (A9):

$$d(\tau_1) = min\big(C_{tr} * h, max\,((0, ((d_s, d_s + d_b), C_1 * h)\,)\,), d_{tr})\big)\ , \tag{A10}$$

Here, the damage parameter $C_{tr}$ describes the upper limit of $d(\tau_1)$ as a fraction of the ice thickness (with the parameter ranging from 0 to 1), and $C_1$ is equal to or less than $C_{tr}$.

**Appendix B: Evolution of the Thwaites Glacier basin by a snapshot in 2100**

In Group 2, 18 samples of the parameters $C_1$ and $C_{tr}$, which represent a very high damage strength, triggered a model collapse

before 2300 (Table A1). Here, we grouped these experiments into Group 2 extreme experiments (G2$_{ext}$) (Fig. A5a). These higher damage strengths in G2$_{ext}$ averagely resulted in a contribution to global mean sea level rise of $7.1 \pm 2.8$ cm by 2100 (the dark red line and its hatched area in Fig. A5b), which is eight times of the mean prediction from the simulations of Group 1. In the simulation with the highest damage strength in G2$_{ext}$, the damage fraction increased from 0.4 at the grounding-line position to 0.7 at the ice front and shear margin of the TG basin in the year 2100 (Figs. A6a). The grounding line retreated by

128 km inland from its position in the year 2020 over a period of only 80 years. The mean annual retreat rate is more than three times of the mean retreat rate simulated by Group 2 experiments and even more than five times of the mean retreat rate simulated by Group 1 experiments (Figs. A6b–A6c). Moreover, the simulated grounding line of the experiment with the highest damage strength in G2$_{ext}$ retreats to a retrograde-slope bed along the central flowline profile in the year 2100, indicating a high potential to retreat further toward inland due to the impact of ice sheet collapse.







**Table A1.** Summary of the damage sensitivity experiments and two control experiments performed at the TG basin under constant present-day conditions. The values of parameters $C_1$ and $C_{tr}$ of the 43 simulations considering the damage processes are produced using Latin hypercube sampling in their parameters space.

| Scenarios | ID | Damage parameters | | Forward simulation type | | RMSEs over 1990-2020 | | |
|---|---|---|---|---|---|---|---|---|
| | | $C_1$ | $C_{tr}$ | Historical simulation | Extended simulation | RMSE (whole basin) | RMSE (floating ice) | RMSE (grounded ice) |
| Ctrl deactivated damage processes | | - | - | 1990-2020 | 2300 | 190.9 | 745.2 | 95.6 |
| Ctrl_cal deactivated damage processes & satellite-observed mass balance calibrated (Otosaka et al., 2023) | | - | - | 1990-2020 | 2300 | 181.3 | 752.7 | 71.0 |
| Group 1 damage processes & SLC and net mass balance within the range of observational estimates ± 2 s.d. (0.24 ± 0.08 cm and -46.1 ± 14.4 Gt a⁻¹) in the historical simulation (Shepherd et al., 2019) | 1 | 0.0576 | 0.7712 | 1990-2020 | 2300 | 327.7 | 1394.3 | 67.1 |
| | 2 | 0.0585 | 0.5215 | 1990-2020 | 2300 | 221.1 | 928.1 | 62.5 |
| | 3 | 0.0657 | 0.3400 | 1990-2020 | 2300 | 196.8 | 815.0 | 63.6 |
| | 4 | 0.0806 | 0.6590 | 1990-2020 | 2300 | 271.4 | 1137.4 | 65.4 |
| | 5 | 0.0838 | 0.5067 | 1990-2020 | 2300 | 233.0 | 974.2 | 63.4 |
| | 6 | 0.0909 | 0.3521 | 1990-2020 | 2300 | 194.8 | 805.4 | 60.4 |
| | 7 | 0.0951 | 0.5530 | 1990-2020 | 2300 | 249.4 | 1041.0 | 63.9 |
| | 8 | 0.1014 | 0.3265 | 1990-2020 | 2300 | 192.3 | 794.2 | 60.2 |
| | 9 | 0.1297 | 0.3007 | 1990-2020 | 2300 | 192.3 | 791.1 | 59.2 |
| | 10 | 0.1385 | 0.4258 | 1990-2020 | 2300 | 234.3 | 967.1 | 62.8 |
| | 11 | 0.1399 | 0.2846 | 1990-2020 | 2300 | 189.4 | 776.2 | 59.7 |
| | 12 | 0.1780 | 0.2613 | 1990-2020 | 2300 | 185.9 | 760.0 | 60.2 |
| | 13 | 0.1819 | 0.2600 | 1990-2020 | 2300 | 185.6 | 757.8 | 60.3 |
| | 14 | 0.1932 | 0.2591 | 1990-2020 | 2300 | 185.2 | 756.2 | 60.2 |
| | 15 | 0.2255 | 0.2588 | 1990-2020 | 2300 | 184.9 | 754.4 | 59.9 |
| Group 2: damage processes & SLC and net mass balance outside the range of observational estimates ± 2 s.d. in the historical simulation | 1 | 0.0174 | 0.9257 | 1990-2020 | 2300 | 178.8 | 709.7 | 77.6 |
| | 2 | 0.0249 | 0.1666 | 1990-2020 | 2300 | 181.1 | 705.8 | 87.3 |
| | 3 | 0.0308 | 0.9861 | 1990-2020 | 2300 | 606.6 | 2468.7 | 77.8 |
| | 4 | 0.0429 | 0.3302 | 1990-2020 | 2300 | 193.2 | 795.9 | 67.3 |
| | 5 | 0.0682 | 0.9409 | 1990-2020 | 2273 | 560.8 | 2217 | 106.7 |
| | 6 | 0.0778 | 0.1783 | 1990-2020 | 2300 | 184.1 | 755.1 | 64.8 |
| | 7 | 0.1232 | 0.9702 | 1990-2020 | 2145 | 1092 | 3962.9 | 260.8 |
| | 8 | 0.1381 | 0.8655 | 1990-2020 | 2164 | 554.8 | 2134.1 | 128.5 |
| | 9 | 0.1486 | 0.6384 | 1990-2020 | 2286 | 489.8 | 1980.6 | 104.6 |
| | 10 | 0.1579 | 0.5134 | 1990-2020 | 2300 | 270 | 1114.8 | 63 |
| | 11 | 0.1759 | 0.8237 | 1990-2020 | 2160 | 427.3 | 1603 | 126.6 |
| | 12 | 0.1807 | 0.4473 | 1990-2020 | 2300 | 253.3 | 1043.6 | 63.1 |





| | | | | | | | |
|---|---|---|---|---|---|---|---|
| 13 | 0.1911 | 0.6941 | 1990-2020 | 2189 | 404.2 | 1586.3 | 102.6 |
| 14 | 0.2267 | 0.7377 | 1990-2020 | 2155 | 403 | 1508.7 | 123.9 |
| 15 | 0.2335 | 0.3962 | 1990-2020 | 2300 | 246.7 | 1015 | 62.9 |
| 16 | 0.2411 | 0.4244 | 1990-2020 | 2300 | 301.7 | 1246.1 | 67.3 |
| 17 | 0.2604 | 0.467 | 1990-2020 | 2284 | 512.1 | 2076.9 | 112.1 |
| 18 | 0.2812 | 0.3218 | 1990-2020 | 2300 | 212.7 | 868.6 | 62 |
| 19 | 0.3068 | 0.648 | 1990-2020 | 2146 | 359.8 | 1334.9 | 124.1 |
| 20 | 0.3497 | 0.6175 | 1990-2020 | 2126 | 552.9 | 2057.9 | 165.6 |
| 21 | 0.3674 | 0.6608 | 1990-2020 | 2101 | 1186.3 | 4249.6 | 301.6 |
| 22 | 0.3789 | 0.4358 | 1990-2020 | 2273 | 526.8 | 2140.1 | 118.2 |
| 23 | 0.3877 | 0.6057 | 1990-2020 | 2113 | 622.4 | 2291 | 190.8 |
| 24 | 0.4129 | 0.5769 | 1990-2020 | 2116 | 574 | 2140.6 | 183.5 |
| 25 | 0.428 | 0.5242 | 1990-2020 | 2167 | 273.4 | 1018.8 | 109.9 |
| 26 | 0.4538 | 0.5433 | 1990-2020 | 2143 | 289.8 | 1075.1 | 112.1 |
| 27 | 0.4711 | 0.5318 | 1990-2020 | 2142 | 262 | 983.3 | 100.3 |
| 28 | 0.5202 | 0.5657 | 1990-2020 | 2124 | 624 | 2330.4 | 192.7 |

**Table A2.** Summary of the forcing and model calibration and evaluation data used in this study.

| Data type | Study | Period | Value |
|---|---|---|---|
| Present-day SMB and temperature (MARv3.11) | Kittel et al., 2021 | 1995-2014 | – |
| Present-day ocean temperature and salinity | Schmidtko et al., 2014 | 1975-2012 | – |
| Contribution of the TG basin to Sea-level rise | Shepherd et al., 2019 | 1992–2017 | $0.24 \pm 0.08$ cm (mean $\pm$ 2 s.d.) |
| Net mass balance of the TG basin | Shepherd et al., 2019 | 1992–2017 | $-46.1 \pm 14.4$ Gt $a^{-1}$ (mean $\pm$ 2 s.d.) |
| MEaSUREs InSAR-Based Antarctica Ice Velocity Map, Version 2 | Rignot et al., 2017 | 1996–2016 | – |
| Surface elevation change of the Amundsen Sea Embayment | Otosaka et al., 2023 | 1992–2019 | |





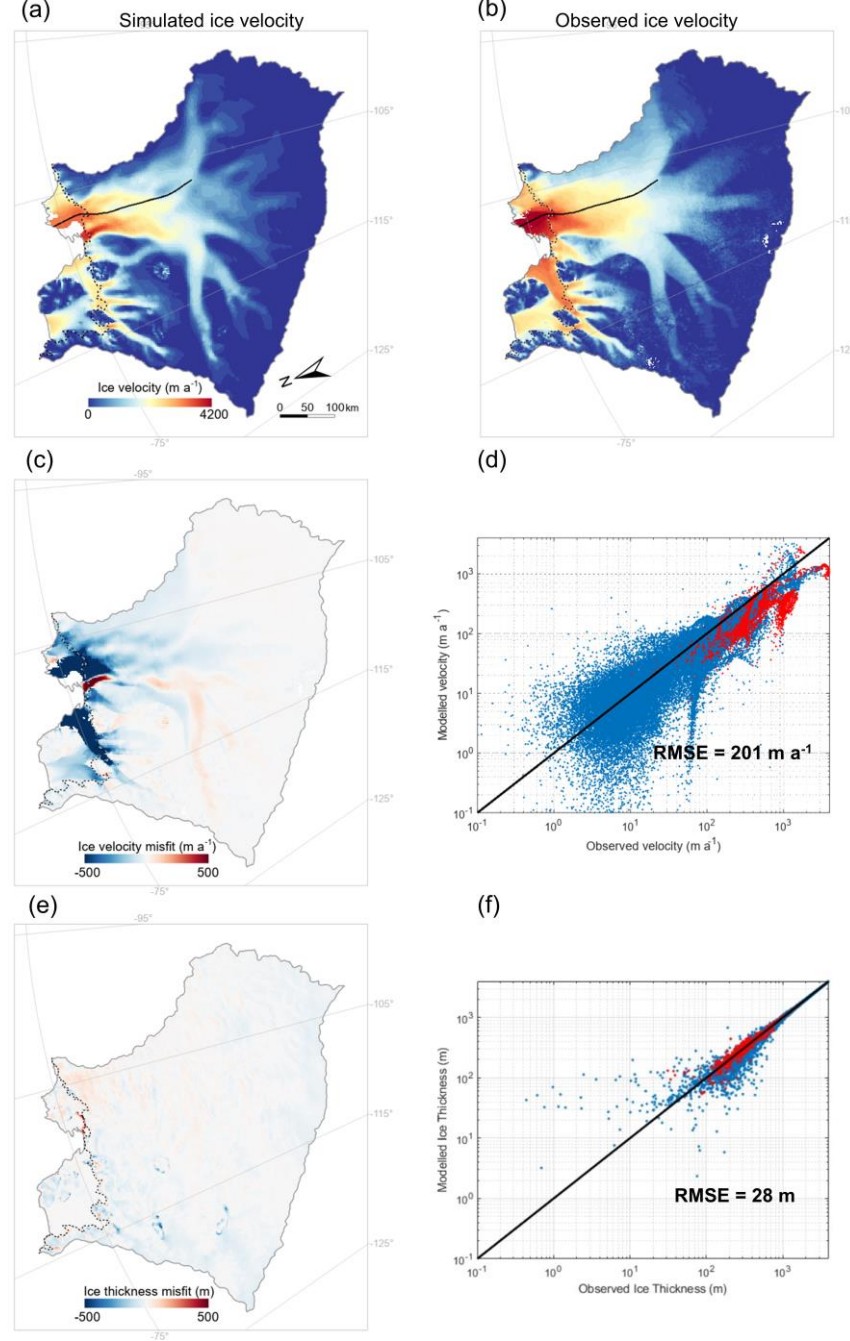


**Figure A1.** Simulated present-day state for the equilibrium initialization obtained with the 1995–2014 atmospheric climatology from MARv3.11 (Kittel et al., 2021). (a) Simulated ice velocity; (b) observed velocity (Rignot et al., 2017); (c) simulated minus observed ice velocity; (d) point-by-point scatter plots of simulated and observed ice sheet (blue) and ice shelf (red) velocities. The black curve is the flowline of Thwaites Glacier derived from the Antarctic surface flowline dataset developed by Liu et al. (2015). The black and gray dashed lines are observed (Gardner et al., 2018) and simulated grounding lines, respectively.



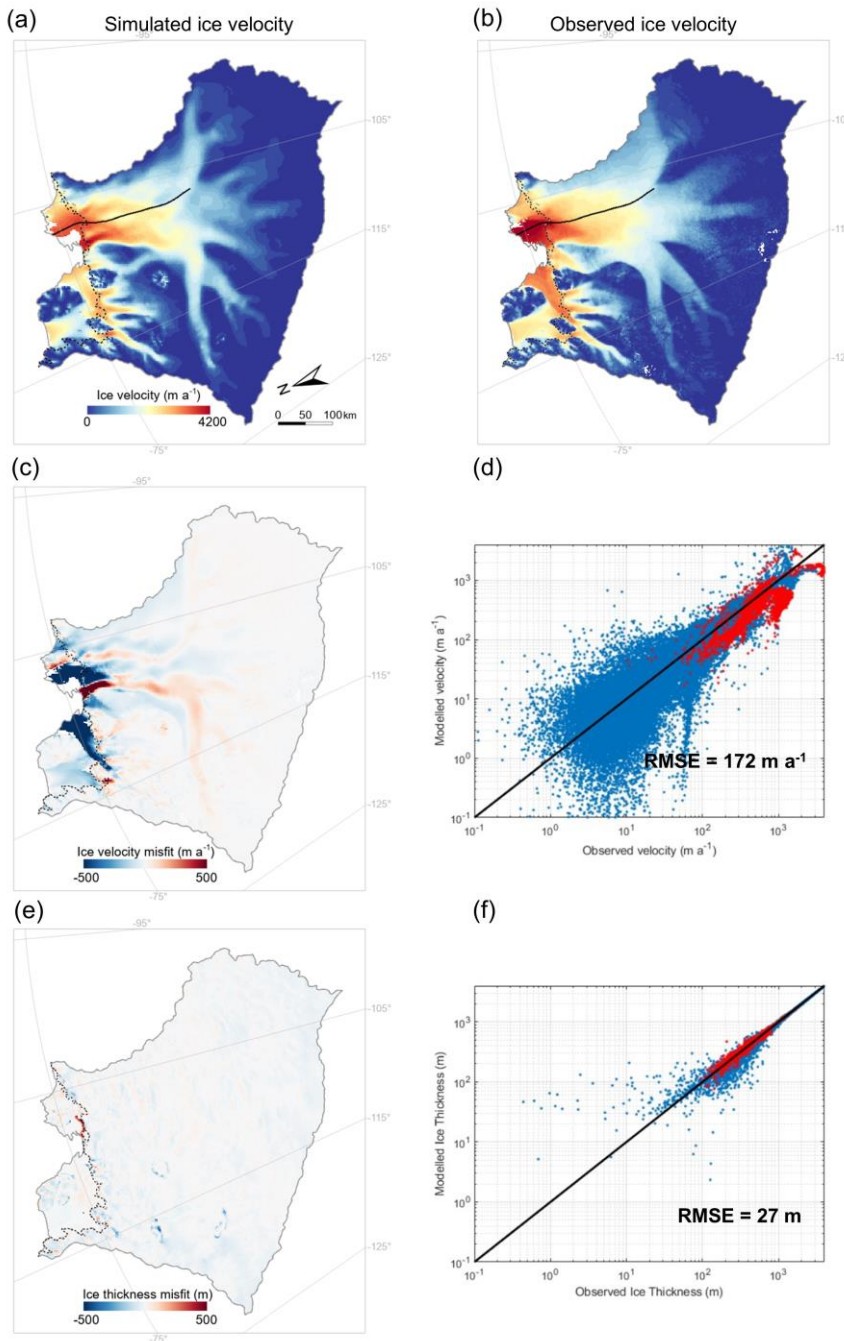

**Figure A2.** Simulated present-day state, same as Figure A1 but with mass balance correction using surface elevation change of the Amundsen Sea Embayment over the period 1992–2019 (Otosaka et al., 2023). (a) Simulated ice velocity; (b) observed velocity (Rignot et al., 2017); (c) simulated minus observed ice velocity; (d) point-by-point scatter plots of simulated and observed ice sheet (blue) and ice shelf (red) velocities. The black curve is the flowline of Thwaites Glacier derived from the Antarctic surface flowline dataset developed by Liu et al. (2015). The black and gray dashed lines are observed (Gardner et al., 2018) and simulated grounding lines, respectively.



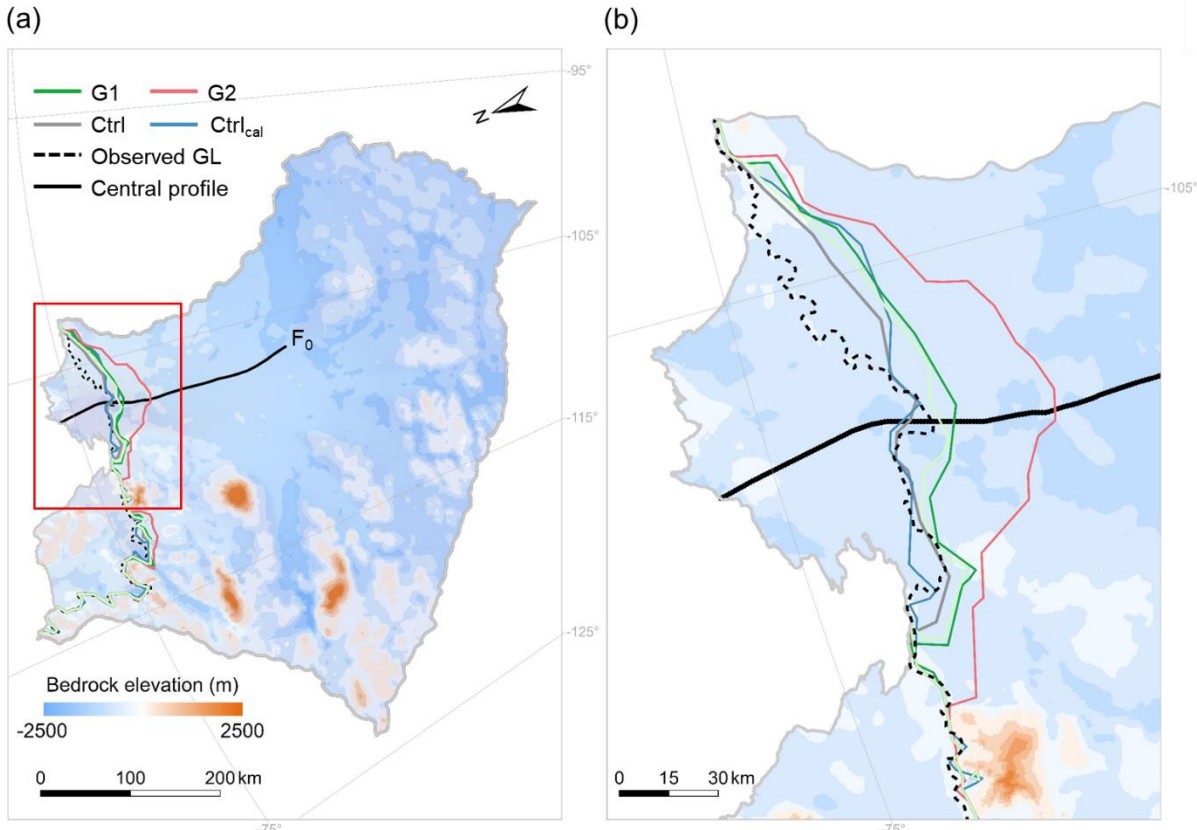

**Figure A3.** Spatial pattern of the grounding-line position in the TG basin over the historical period 1990–2020 under different damage strengths. (a) Evolution of the grounding-line position within the TG basin and (b) an enlarged view of the red box in Figure (a). The light green and dark green lines represent the experiments with the least and the most grounding-line retreat in Group 1, respectively, and also correspond to the experiments with the lowest and highest damage strength in Group 1. The red line represents the experiment with the most grounding-line retreat in Group 2, and also corresponds to the experiment with the highest damage strength in Group 2 over the historical period 1990–2020. The black dashed line presents the observed grounding-line position (Gardner et al., 2018). The blue and grey lines present simulated grounding-line positions of the $Ctrl_{cal}$ and $Ctrl$ experiments, respectively. The background figure in (a) is the observed bedrock elevation of the TG basin derived from BedMachine v2 data (Morlighem et al., 2020).





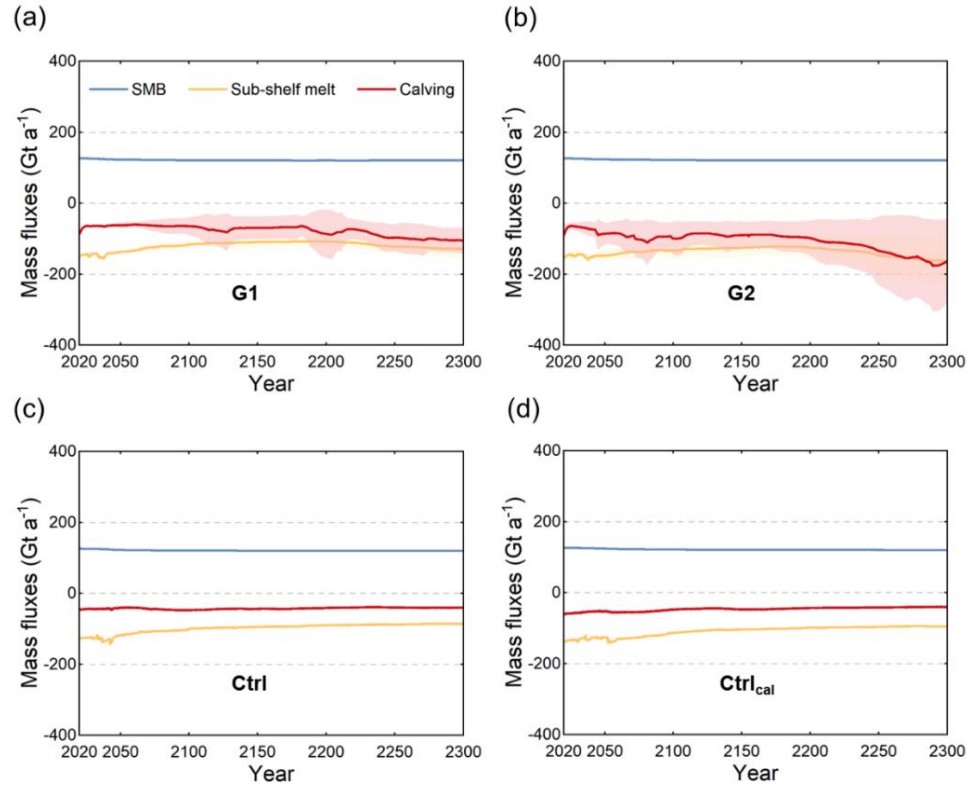

**Figure A4.** Evolution of the mass balance components including surface mass balance (SMB), the sub-shelf melt fluxes, and dynamic ice loss (i.e. the calving fluxes) under (a) Group 1, (b) Group 2, (c) Ctrl, and (d) Ctrl$_{cal}$ experiments over the projection period 2020–2300. Solid line represents mean, hatched area represents ensemble standard deviation.

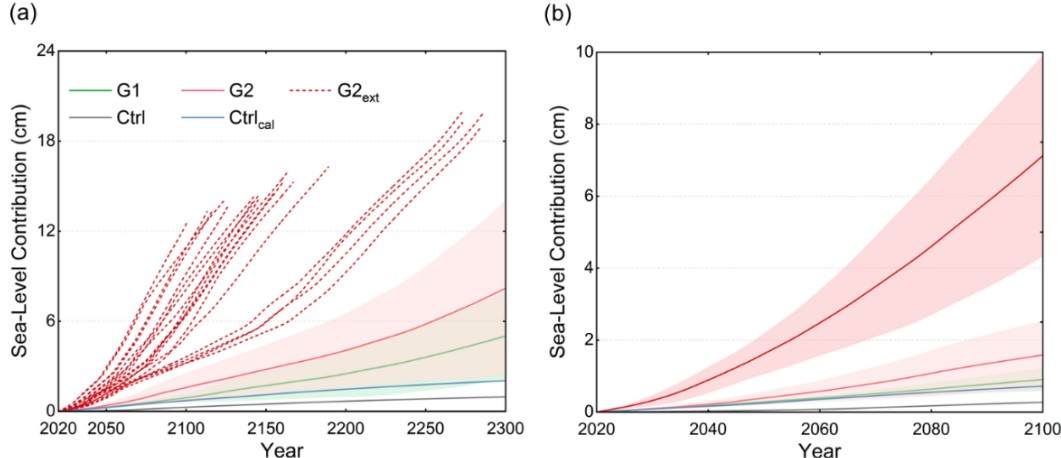

**Figure A5.** Evolution of the contribution to global mean sea-level rise of the TG basin over (a) the projection period 2020–2300, and (b) with a focus on the period 2020–2100 under constant present-day conditions. The dashed red lines in (a) represent experiments with higher damage strengths that triggered a model collapse before 2300 and were grouped into Group 2 extreme experiments (G2$_{ext}$). Solid line represents mean, hatched area represents ensemble standard deviation.



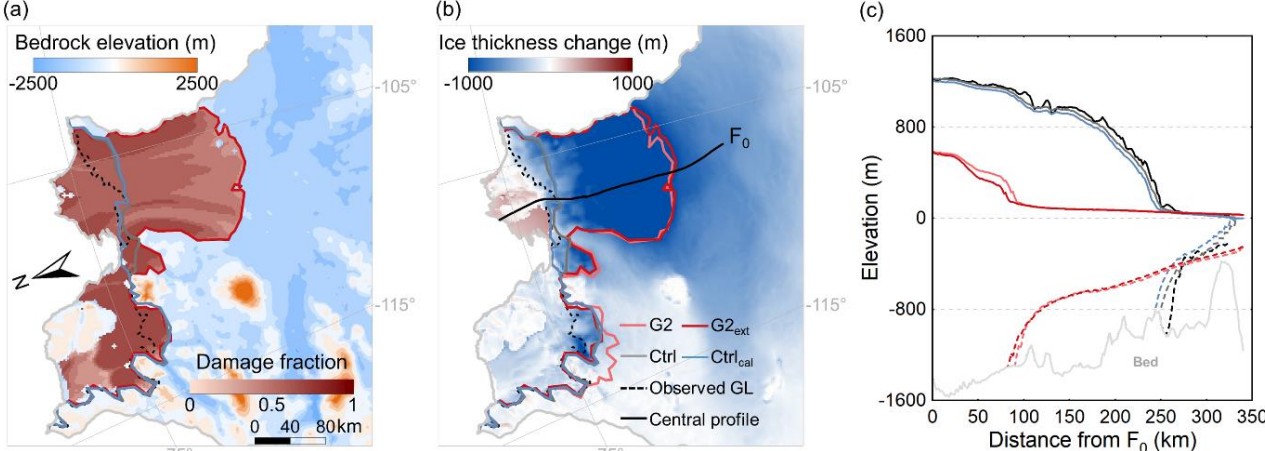

**Figure A6.** (a) Damage field, (b) ice thickness change, and (c) ice geometry along the central profile of the simulation with the highest damage strength in $G2_{ext}$ in the year 2100. The dark (light) red lines represent the spatial pattern of the simulated grounding-line position and the ice geometry along the central profile of the simulation with the highest damage strength in $G2_{ext}$ (G2) in the year 2100 (2300). The black dashed line presents the observed grounding-line position (Gardner et al., 2018). The blue and grey lines present simulated grounding-line positions of the $Ctrl_{cal}$ and Ctrl experiments, respectively. The background figure in (a) is the observed bedrock elevation of the TG basin derived from BedMachine v2 data (Morlighem et al., 2020).

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
