# Peer review of "Damage strength increases ice mass loss from Thwaites Glacier, Antarctica"

_EGUsphere, 2024_

## Referee Comment (RC1)

**A review of "Damage strength increases ice mass loss from Thwaites Glacier, Antarctica" by Y. Li et al.**

This manuscript aims to address an important question in ice sheet modeling studies, i.e., if and how much the ice damage affects the ice flow in some vunerable regions like WAIS. Thus, it is no doubt a valuable study and lies perfectly in the scope of TC.

**General remarks:**

- Despite the damage method has been used in some previous numerical studies, it is still necessary to compare the modeled damage field with observed crevass and rift images - I think it is critical to convince us how much we can trust the damage model results.
- If the authors aim to give a plausible projection of GMSL contribution from Thwaites Glacier, then it is necessary to use CMIP forcing data, from both atmosphere and ocean.
- And more details of forcing data and model configurations are needed. See the following details:

**Details:**

L10: damage is a result (or metric) of crevasses, not the reason.

L17: GMSL instead of sea-level rise

L21: again, damage is the result of rifts and crevasses

L29-39: the review has not included other studies, e.g., Duddu et al. (2020) and Kachuck et al. (2022), and probably many others, a big improvemt of this paragraph is highly necessary.

L40-54: It would be very helpful if you can provide an image showing the crevass distribution across Thwaites Glacier along with the current Fig 1.

L55: Here I think there probably lacks a paragraph describing what diagnostic and prognostic modeling studies we currently have, and what kind of problems in those studies have by not including the damage mechanics in their models, before you move on to this paragraph introducing your solutions.

L76: "zero-stress assumption" might be better

L79: do not understand how you get d1(tau1) even after looking at Appendix 1. Intuitively, it looks like to be min((ds+db), C1*h), i.e., the min value between the total crevass depth and the limit you set. Can you provide more explanations?

L80: remove the comma after where

L83: remove extra () for d(tau1)

L84: I need more details of d1(tau1) to understand this equation

L85: remove the comma after where

L92: change "steady state" to "steady-state", same for other places

L102-109: have you compared the modeled damage results to the observed crevasse distribution from satellite image? I think this is also important.

L110-116: What are the forcings for the experiments Ctrl and Ctrl_cal? Do you calibrate the forcing data for Ctrl_cal in order to reproduce the historical trend of ice mass change? If so, how do you do the calibration? What is the time span for the historical runs?

L119: what is the RMSE for grounding line position?

L120: I do not follow the sentence "At the start of the historical run, the present-day SMB is reinstated without the additional mass-change term". Can you explain it a bit more?

L124: So no CMIP projection forcing data? Then we should be careful to conclude a GMSL contribution from this study, as it is more like a comparison (damage v.s. no damage) study.

L129: how do you do with the basal melt rates for previously grounded the regions after they become floating as GL retreats? Do you couple the PICO model with the ice sheet model?

L141: 43-member?

Section 3.1: There are something I do not understand in this part. For Ctrl_cal, you can actually calibrate the forcing and let the modeled and observed mass change match each other, even you do not turn on the damage mechanism, correct? But from Fig 2a, clearly there is still some disagreement between Ctrl_cal and the observations. Why is that? For G1 and G2, basically what you do is damage parameter calibration, and you can find some parameter combinations that can give a good model output. But how can you tell the difference of model and observations is not from the bias of the forcing data you use, but is due to the damage mechanism? That is the point I am still confused. That is another reason that I think a comparison between modeled damage value and obserbed crevasse distribution is necessary. The current form of Fig 2 needs improvements too. It is hard to tell those curves for G1 and G2. I would suggest to keep only several curves that are close to observations, and put the whole ensemble somewhere in the Appendix.

L176-192 and Fig 3: So this part explains again my concern for Section 3.1. The RMSE of Ctrl_cal is even smaller than G1-G15. Does that mean we can calibrate the forcing data, e.g., basal melt rates, to get a better hindcast modeling result than tuning the damage parameters, or can we say that forcing is more important than damage? In Fig 3, I would also like to see the comparison of modeled and observed ice thickness data, which is a also very important information, or you might consider to add an additional figure for thickness.

Section 3.2: I'll hold my opinions for this section for now, as I do not see much information about SMB and basal melt forcings, e.g., if you use a coupling scheme between ice sheet and ocean model or you use a some parameterization approach. Before I have these information, I can't tell how meanful the projection numbers are in this section.

Figure 6: you do have the current damage field. Then I think you probably want to compare them with some satellite images of crevasses. I think it is important for us to understand if the damage method you use is valid or not.

**References:**

Kachuck SB, Whitcomb M, Bassis JN, Martin DF, Price SF. Simulating ice-shelf extent using damage mechanics. Journal of Glaciology. 2022;68(271):987-998. doi:10.1017/jog.2022.12,

Duddu R, Jiménez S, Bassis J. A non-local continuum poro-damage mechanics model for hydrofracturing of surface crevasses in grounded glaciers. Journal of Glaciology. 2020;66(257):415-429. doi:10.1017/jog.2020.16

---

## Author Response (AR1)

**egusphere-2024-2916**

**Responses to comments from Reviewer Tong Zhang**

Damage intensity increases ice mass loss from Thwaites Glacier, Antarctica

We thank the editor and the two reviewers for their constructive feedback, which has helped us improve the manuscript. In response to the comments, we have thoroughly revised the manuscript. The key updates are the following:

- We have improved the writing, formulation and structure of the manuscript to enhance clarity.
- Appendix A has been removed and merged into section 2.1, where the model formulation has been extensively revised.
- To provide greater clarity on our methodology, we have introduced separate sections for the simulation protocol (Section 2.2) and the initialization (Section 2.3).
- Appendix B has been removed and integrated into the results section, which has been thoroughly revised to improve readability.
- A comparison of the modeled damage pattern and observations has been added (see Figure 5 in the revised manuscript).

Further details on these revisions are provided in our responses below. In the following, we use "**bold text**" for the reviewer's comments, "regular" text for our responses, and "*italic*" for text extracted from the manuscript.

**General remarks:**

**This manuscript aims to address an important question in ice sheet modeling studies, i.e., if and how much the ice damage affects the ice flow in some vulnerable regions like WAIS. Thus, it is no doubt a valuable study and lies perfectly in the scope of TC.**

**Despite the damage method has been used in some previous numerical studies, it is still necessary to compare the modeled damage field with observed crevasse and rift images – I think it is critical to convince us how much we can trust the damage model results.**

Response:

Thank you for your comments. As suggested, we have compared our simulated damage fields with observed crevasse distribution and added the relevant description in **Section 3.1** of the revised manuscript. We have also included a new figure (Figure 5 in the revised manuscript) showing the crevasse distribution across the ice shelf regions of the Thwaites Glacier basin, derived from Landsat-8 satellite images (December 2020), alongside our present-day simulated damage fields. Please see the following responses to specific comments and the revised manuscript for further details.

**If the authors aim to give a plausible projection of GMSL contribution from Thwaites Glacier, then it is necessary to use CMIP forcing data, from both atmosphere and ocean.**

Response:

Thank you for your comment.

We would like to emphasize that this study does not aim to produce sea-level projections. Instead, we focus on testing the influence of ice damage on the Thwaites Glacier basin under constant present-day climate conditions. Our sensitivity experiments allow us to quantify TG's mass loss response to the damage feedback mechanism.

To clarify this in the manuscript, we have revised our presentation of the results, shifting the focus away from GMSL and instead emphasizing relative mass changes. For details, please see the following responses as well as the revised manuscript.

**And more details of forcing data and model configurations are needed. See the following details:**

Response:

Thank you for your comment. We have added more details on the forcing data and model configurations, as suggested. Especially, to provide greater clarity on our methodology, we have introduced separate sections for the simulation protocol (Section 2.2) and the initialization (Section 2.3). Please see the following responses and the revised manuscript for further details.

**Details:**

**1) L10: damage is a result (or metric) of crevasses, not the reason.**

Response:

Thank you for your comment. We revised this sentence as follows:

(Line 9–10 in the revised manuscript without tracks): "*...Ice damage, which results from the formation and development of crevasses on glaciers, plays a critical role in ice-shelf stability, grounding-line retreat, and subsequent sea-level rise....*".

**2) L17: GMSL instead of sea-level rise**

Response:

Thank you. As mentioned above, we have revised our presentation of the results, shifting the focus away from GMSL and instead emphasizing absolute and relative mass changes. We revised this sentence as follows:

(Line 15–17): "*...When extending simulations to the year 2300, we show that*

*accounting for ice damage results in more than twice the ice mass loss compared to simulations that neglect ice damage mechanics...."*

**3) L21: again, damage is the result of rifts and crevasses**

Response:

Thank you for your comment. We revised this sentence as follows:

(Line 20–21): "*...The weakening of ice due to the formation of large-scale crevasses and rifts, known as damage, is gaining attention due to its impact on glacier and ice sheet evolution in a warming climate...."*

**4) L29-39: the review has not included other studies, e.g., Duddu et al. (2020) and Kachuck et al. (2022), and probably many others, a big improvement of this paragraph is highly necessary.**

Response:

Thank you for your feedback. We have carefully revised the introduction and incorporated additional relevant studies (e.g., Duddu et al., 2020; Kachuck et al., 2022, Huth et al., 2021, 2023; Ranganathan et al., 2024).

Duddu, R., Jiménez, S., and Bassis, J.: A non-local continuum poro-damage mechanics model for hydrofracturing of surface crevasses in grounded glaciers, Journal of Glaciology, 66(257), 415-429, doi:10.1017/jog.2020.16, 2020.

Kachuck, S. B., Whitcomb, M., Bassis, J. N., Martin, D. F., and Price, S. F.: Simulating ice-shelf extent using damage mechanics, Journal of Glaciology, 68(271), 987-998, doi:10.1017/jog.2022.12, 2022.

Huth, A., Duddu, R., and Smith, B.: A generalized interpolation material point method for shallow ice shelves. 2: Anisotropic nonlocal damage mechanics and rift propagation, Journal of Advances in Modeling Earth Systems, 13(8), e2020MS002292, doi:10.1029/2020MS002292, 2021.

Huth, A., Duddu, R., Smith, B., and Sergienko, O.: Simulating the processes controlling ice-shelf rift paths using damage mechanics, Journal of Glaciology, 69(278), 1915 – 1928, doi: 10.1017/jog.2023.71, 2023.

Ranganathan, M., Robel, A. A., Huth, A., and Duddu, R.: Glacier damage evolution over ice flow timescales, EGUsphere [preprint], https://doi.org/10.5194/egusphere-2024-1850, 2024.

**5) L40-54: It would be very helpful if you can provide an image showing the crevasse distribution across Thwaites Glacier along with the current Fig 1.**

Response:

Thank you for the suggestion. We have revised Figure 1 by adding Figures 1c and 1d showing the crevasses distribution across the ice shelf regions of the Thwaites

Glacier basin, based on Landsat-8 satellite images derived in December, 2020.

[Figure]

**Figure 1.** *Bedrock elevation and ice velocity in the TG basin. (a) Observed bedrock elevation of the TG basin based on BedMachine v2 data (Morlighem et al., 2020) and (b) observed ice velocity of the TG basin based on Making Earth System Data Records for Use in Research Environments (MEaSUREs) InSAR-Based Antarctica Ice Velocity Map, Version 2 (Rignot et al., 2017) overlaid on the Landsat Image Mosaic of Antarctica (LIMA; Bindschadler et al., 2008). The solid black curve is the central flowline profile stemming from the Antarctic surface flowline dataset developed by Liu et al. (2015), which spans 340 km from the inland grounded ice ($F_0$) to the calving front. The dashed black line shows the position of the observed grounding line (Gardner et al., 2018). The inset in panel (a) shows the location of the TG basin in Antarctica. TEIS represents the Thwaites Eastern Ice Shelf and TWGT represents the Thwaites Western Glacier Tongue. The black rectangular insets in panel (a) are panels (c) and (d), which show the crevasses distribution across the ice shelf regions in the TG basin, based on Landsat-8 satellite images*

*acquired in December 2020. The gray line is the basin boundary of the TG basin derived from Zwally et al. (2015).*

**6) L55: Here I think there probably lacks a paragraph describing what diagnostic and prognostic modeling studies we currently have, and what kind of problems in those studies have by not including the damage mechanics in their models, before you move on to this paragraph introducing your solutions.**

Response:

Thank you for your suggestion. We have revised the introduction and added separate paragraphs discussing existing diagnostic and prognostic studies.

(Line 32–59): "*…Several studies have investigated the influence of damage on the behavior of the Antarctic Ice Sheet (AIS). Borstad et al. (2012) applied a large-scale ice dynamical model to invert for damage on the Larsen B Ice Shelf prior to its collapse in 2002. They concluded that calving was triggered by the loss of load-bearing surface area due to fracturing. Albrecht and Levermann (2014) investigated the role of damage in softening ice across several Antarctic ice shelves using a fracture density field derived from observations. Gerli et al. (2023) demonstrated that the vertical propagation of crevasses within ice shelves can instantaneously increase the flux of upstream glaciers. Huth et al. (2021, 2023) integrated a creep damage model into a large-scale shallow-shelf ice flow model to simulate rift propagation leading to the formation of iceberg A68 from the Larsen C Ice Shelf. Damage is facilitated through hydrofracturing, and the combined effect of non-linear viscous rheology and damage processes within ice at water-filled crevasse tips can influence calving dynamics (Duddu et al., 2020). Sun and Gudmundsson (2023) conducted a series of numerical perturbation experiments to show that damage evolution significantly affects ice-shelf velocities and must be accounted for to accurately replicate observed velocity patterns. These studies reveal the interaction between damage processes and observed ice flow dynamics. They have one critical limitation, i.e., being diagnostic, which means that they investigate the instantaneous effect of damage on ice dynamics, but not the evolution of damage when ice thickness is allowed to evolve according to the applied changes. They therefore fail to predict future ice sheet behavior or feedbacks induced by external changes, such as fracture enhancement due to atmospheric or oceanic forcing.*

*Prognostic modeling enables the assessment of ice sheet and ice shelf evolution in response to fracture dynamics. However, most existing studies focus on idealized ice sheet geometries. Sun et al. (2017) coupled a continuum damage mechanics (CDM) model with an ice-sheet model based on the zero-stress Nye approach (Nye, 1957). Applying this model to an idealized ice-sheet geometry (MISMIP+; Cornford et al., 2020), they found that ice damage leads to greater grounding-line retreat compared to simulations without damage. Using the same model, Lhermitte et al. (2020) showed that intensifying damage at a specific location within shear*

*zones triggers widespread propagation and amplification of damage, supporting the hypothesis of a positive feedback mechanism. By integrating a continuum damage mechanics model with necking instability into an ice sheet model, Kachuck et al. (2022) simulated the evolution of the damage field and accurately predicted steady-state extents for a series of idealized, isothermal ice tongues and ice shelves. Similarly, Ranganathan et al. (2024) developed a damage evolution model coupled with a marine-terminating glacier flowline model and showed that damage can enhance mass loss from both grounded and floating ice. However, the results obtained from idealized geometries may not fully translate to the real world conditions, and studies investigating the effects of ice damage on the dynamics of actual glaciers, such as Antarctic glaciers and ice shelves, remain limited.…"*

**7) L76: "zero-stress assumption" might be better**

Response:

Thank you for your suggestion, this has been modified.

**8) L79: do not understand how you get d1(tau1) even after looking at Appendix 1. Intuitively, it looks like to be min((ds+db), C1\*h), i.e., the min value between the total crevass depth and the limit you set. Can you provide more explanations?**

Response:

Apologies for the confusion. The equation of $d_1(\tau_1)$ should be in the following form:

$$d_1(\tau_1) = min(d_s + d_b, C_1 * h)$$

We use the parameter $C_1$ to impose an upper limit to $d_1(\tau_1)$ as a fraction of the ice thickness (with $C_1$ ranging from 0 to 1), preventing an overestimation of crevasse depth in our gridded domain.
We have also corrected the equation and revised section 2.1 accordingly.

**9) L80: remove the comma after where**

Response:

Done.

**10) L83: remove extra () for d(tau1)**

Response:

Done.

**11) L84: I need more details of d1(tau1) to understand this equation**

Response:

Please see the response to **comment L79**.

**12) L85: remove the comma after where**

Response:

Done.

**13) L92: change "steady state" to "steady-state", same for other places**

Response:

Thank you for your comment. In alignment with both this suggestion and parallel feedback from other reviewers, we have revised the relevant sentence as follows:

(Line 159–162): "...*In the damage sensitivity experiments, ice damage is activated from the first timestep of the historical simulation, meaning that the ice sheet is considered undamaged at the start of 1990. Given that this assumption is somewhat idealized, the simulated damage can be interpreted as relative to the initial state. ...*"

**14) L102-109: have you compared the modeled damage results to the observed crevasse distribution from satellite image? I think this is also important.**

Response:

Thank you for your suggestion. Figure 5 of the revised manuscript now allows for a comparison of observed and modeled crevasse distribution. The following statement has been included in the results section:

(Line 276–287): "...*Figure 5a presents the distribution of crevasses observed across the ice shelves of the TG basin, derived from Landsat-8 satellite images taken in December 2020. Our vertically averaged ice damage patterns tend to overestimate damage on the Dotson ice shelf, suggesting the need for a threshold stress parameter to better capture damage initiation. In contrast, ice fracture is underestimated in the Thwaites Western Glacier Tongue, likely due to the stabilizing influence of the Northwest pinning point (Surawy-Stepney et al., 2023). ...*"

[Figure]

*Figure 5. Damage distribution in the TG basin. (a) Observed crevasse distributions across the ice shelves of the TG basin, based on Landsat-8 satellite images acquired in December 2020. Vertically averaged damage fields (i.e., $d(x, y)/h(x, y)$) in the year 2000 and 2020 of the low damage intensity of Group 1 (G1) are shown in (b) and (e); the high damage intensity of G1 in (c) and (f); and the high damage intensity of Group 2 (G2) in (d) and (g). The dashed black line is the observed grounding line (Gardner et al., 2018). The light gray line is the basin boundary of the TG basin derived from Zwally et al. (2015). The dashed gray and blue lines present the initial grounding-line positions of the Ctrl/damage experiments and the $Ctrl_{dhdt}$ experiment, respectively.*

The comparison with observations is also discussed in the Discussion section:

(Line 395–413): "…*Our approach has the benefit of using a physical approach to infer crevasse formation. However, direct comparison with observation remains challenging, since the damage field is highly variable and corresponds to a particular time moment. Our results are highly dependent on the forcing and model uncertainties, which makes a direct comparison unfeasible. Moreover, modeled damage patterns are highly variable across ensemble members. These discrepancies may be explained by the limitations of the damage model. For example, our approach does not account for all mechanisms of damage healing, which may result in an overestimation of damage (Sun et al., 2017). In reality,*

*crevasse healing can occur when shear stress along the flow path decreases notably (Wesche et al., 2013; Benn and Åström, 2018), and dense crevasses near the grounding zone may heal during their advection towards the calving front. However, studies on the process of ice healing are still scarce due to the challenges of monitoring and quantifying this process (Albrecht and Levermann, 2012). Additionally, a vertically-integrated model may not be appropriate for accurately representing crevasse formation mechanisms. The application of threshold stress for damage initiation as well as mechanisms of crevasse healing, such as the accretion of marine ice within basal crevasses, should be explored (Sun et al., 2017). The lack of representation of plastic necking (Bassis and Ma, 2015) also introduces uncertainties in our results. While the comparison of modeled, vertically integrated damage fields with snapshots of surface crevasses is not straightforward, these discrepancies underline the need for further validation and calibration of the damage model. Instead of solely relying on ice sheet mass loss data, future efforts should incorporate observational datasets of crevasse distributions. Moreover, while the simulated historical state of the TG basin is overall consistent with observations, the 1995–2014 mean boundary conditions used to initialize the model and simulate hindcasts for 1990–2020 (Schimdtko et al., 2014; Kittel et al., 2021) do not necessarily reflect the actual imbalance of the ice sheet during that period. ...”*

**15) L110-116: What are the forcings for the experiments Ctrl and Ctrl_cal?**

Response:

All simulations in our study, including experiments Ctrl and $Ctrl_{cal}$ (renamed '$Ctrl_{dhdt}$' in the revised manuscript for clarity) are forced with constant present-day conditions, using the present-day surface mass balance and temperature obtained from the polar regional climate model MARv3.11 (Kittel et al., 2021) and present-day ocean temperature and salinity derived from data provided by Schmidtko et al. (2014).

This has been clarified in the revised manuscript.

**Do you calibrate the forcing data for Ctrl_cal in order to reproduce the historical trend of ice mass change? If so, how do you do the calibration?**

Response:

To reproduce the dynamic disequilibrium observed during the historical period in the '$Ctrl_{cal}$' experiment (renamed '$Ctrl_{dhdt}$' in the revised manuscript for clarity), we apply the initialization method described in van den Akker et al. (2025). Specifically, the initial state of the $Ctrl_{dhdt}$ experiment is obtained by adding a 'correction term' - equal to minus the observed mass change rates - to the present-day surface mass balance during the transient nudging procedure. This ensures that by the time the nudging procedure has achieved a constant geometry, the model has been trained to produce ice fluxes that closely match observations. In other

words, the ice sheet model is 'trained' to equilibrate toward a state where observed mass change rates are implicitly accounted for.

We have also revised the related sentences to make it clearer.

(Line 163–168): "…*To reproduce the dynamic disequilibrium observed during the historical period, we apply the initialization method of van den Akker et al. (2025). Specifically, the initial state of the Ctrl$_{dhdt}$ experiment is obtained by adding a 'correction term' – equal to minus the observed mass change rates (taken from Bevan et al., 2023) – to the present-day surface mass balance (Kittel et al., 2021) during the transient nudging procedure. This ensures that, by the time the nudging procedure has achieved a steady geometry, the model has been trained to produce ice fluxes that closely match observations. In other words, the ice sheet model is 'trained' to equilibrate toward a state that implicitly accounts for observed mass change rates.…*"

van den Akker, T., Lipscomb, W. H., Leguy, G. R., Bernales, J., Berends, C., van de Berg, W. J., and van de Wal, R. S. W.: Present-day mass loss rates are a precursor for West Antarctic Ice Sheet collapse, The Cryosphere, 19, 283–301, doi:10.5194/tc-19-283-2025, 2025.

**What is the time span for the historical runs?**

Response:

The time span of the historical simulations is 30 years from 1990 to 2020. We have clarified that in the revised manuscript.

**16) L119: what is the RMSE for grounding line position?**

Response:

We calculated the mean distance between the modeled grounding-line positions and the observed grounding-line position based on the "open-ended box" method proposed by Moon and Joughin (2008). We have clarified this in the revised manuscript.

(Line 176–178): "…*In addition, we estimate the mean distance between the modeled and observed grounding-line position using the "open-ended box" approach of Moon and Joughin (2008).…*"

We have also added the information about modeled grounding-line position for the Ctrl and the Ctrl$_{dhdt}$ experiments into the revised manuscript:

(Line 181–182): "…*The modeled grounding-line position of the TG basin is in good agreement with observations (Gardner et al., 2018), with an average offset of 1.3 km.…*"

(Line 184–185): "…*The modeled grounding-line position also closely aligns with observations, with an average offset of 2.3 km.…*"

17) **L120: I do not follow the sentence "At the start of the historical run, the present-day SMB is reinstated without the additional mass-change term". Can you explain it a bit more?**

Response:

As developed in the response to comment 15), the initial state of the $Ctrl_{dhdt}$ experiment is obtained by adding a 'correction term' - equal to minus the observed mass change rates - to the present-day surface mass balance during the transient nudging procedure. This correction term is removed from the surface mass balance at the start of the historical run. As a result, the ice will start to thin/thicken at (almost) exactly the observed rates. That is, the model will reproduce per construct the observed mass-balance rates as a drift.

18) **L124: So no CMIP projection forcing data? Then we should be careful to conclude a GMSL contribution from this study, as it is more like a comparison (damage v.s. no damage) study.**

Response:

Indeed, it is important to underline that this study does not aim to produce sea-level projections. Instead, we focus on testing the influence of ice damage on the Thwaites Glacier basin under constant present-day climate conditions. Our sensitivity experiments thus allow us to quantify TG's mass loss response to the damage feedback mechanism.

To clarify this, we have shifted the focus away from GMSL, instead emphasizing relative mass changes, throughout the manuscript.

19) **L129: how do you do with the basal melt rates for previously grounded the regions after they become floating as GL retreats? Do you couple the PICO model with the ice sheet model?**

Response:

Yes, the basal melting underneath the floating ice shelves is estimated at each time step using the PICO model of Reese et al. (2018).

20) **L141: 43-member?**

Response:

Correct, thank you for spotting this.

21) **Section 3.1: There are something I do not understand in this part. For Ctrl_cal, you can actually calibrate the forcing and let the modeled and observed mass**

**change match each other, even you do not turn on the damage mechanism, correct? But from Fig 2a, clearly there is still some disagreement between Ctrl_cal and the observations. Why is that?**

Response:

Thank you for your question. By integrating the observed mass change rates during the initialization procedure (Ctrl$_{dhdt}$), the modeled and observed mass changes can indeed be matched, even without activating the damage mechanism.

First, it is important to clarify that the observational data used for the initializing procedure of the Ctrl$_{dhdt}$ experiment and those used for validation in Figure 2 are not the same. For the model initialization, we use 2D satellite-based data of present-day ice mass change rates in the TG basin (Bevan et al., 2023) to correct the present-day surface mass balance. For the validation in Figure 2, we use satellite-derived observations of the sea-level contribution between 1992 and 2017, spatially-aggregated over the basin, from Shepherd et al. (2019).

Your comment likely stems from how the observations were initially represented in the figure, which was not optimal. We now have adjusted the figure by representing the observations as an error bar on the right side of the plot. This hopefully now better shows that the Ctrl$_{dhdt}$ captures the observational trends well.

**For G1 and G2, basically what you do is damage parameter calibration, and you can find some parameter combinations that can give a good model output. But how can you tell the difference of model and observations is not from the bias of the forcing data you use, but is due to the damage mechanism? That is the point I am still confused.**

Response:

Thank you for your comment. To quantify the impact of the damage mechanism, we conducted two baseline experiments (Ctrl and Ctrl$_{dhdt}$) excluding damage throughout both the historical and projection simulations. All experiments (with or without damage) are forced with constant present-day climate conditions.

The Ctrl experiment starts from the same initial state as the 43-member ensemble that includes damage. Comparing these simulations therefore allows us to assess the influence of damage on the evolution of the TG basin. Our historical simulation results show that when damage is considered (G1), the simulated sea-level contribution (SLC) and net mass balance align well with observations, whereas the Ctrl experiment fails to reproduce the observations accurately. This indicates that the mass balance trend of G1 is induced by the damage feedback rather than by a bias in the forcing. That said, we acknowledge that no model is perfect -- a simulation may match the observations for the wrong reason, just as it may diverge from them despite incorporating relevant physics (here, damage parameters). To account for this, we deliberately adopted a flexible calibration approach, including G1 simulations that fall within ± twice the observational error.

Note that the Ctrl$_{dhdt}$ experiment does match the observations, but because its initial

state was explicitly adjusted to equilibrate toward a state that implicitly accounts for observed mass change rates.

**That is another reason that I think a comparison between modeled damage value and observed crevasse distribution is necessary.**

Response:

Agreed. Please see our response to **comment 14)** above.

**The current form of Fig 2 needs improvements too. It is hard to tell those curves for G1 and G2. I would suggest to keep only several curves that are close to observations, and put the whole ensemble somewhere in the Appendix.**

Response:

Thank you for your comment. To improve the clarity of Figure 2, we have added hatched areas showing the range (i.e., the spread between the minimum and maximum values) for the G1 and G2 ensembles (see below).

[Figure]

***Figure 2.*** *Simulated change trends of ice mass balance and grounding-line position in the TG basin under different damage intensities over the period 1990–2020. (a) the simulated contribution of ice mass loss in the TG basin to sea level; (b) the net mass balance (considering volume above flotation only, i.e., the rate of mass change contributing to sea-level rise) in the TG basin; (c) the geometry profiles along the central flowline profile (solid black line in Fig.1) and the simulated (dashed red and green lines) grounding-line positions; (d) the simulated ice velocity along the central flowline profile. RMSEs between the simulated and*

*observed ice velocity under different parameter combinations of $C_1$ and $C_{tr}$ in (e) Group 1 and (f) Group 2. The dark red and green lines in panels (a)–(d) represent the mean, and the hatched area represents the ensemble range, i.e., spread between maximum/minimum values. The black lines and shaded areas in panels (a) and (b) represent the observed mean value ± 1 standard deviation (Shepherd et al., 2019). The gray line represents the simulation result of the model that ignored ice damage processes and did not integrate satellite-based observations of present-day mass-change rates to constrain the model initialization (Ctrl experiment), and the blue line represents the simulation result of the model that ignored ice damage processes but integrated satellite-based observations to constrain the model initialization (Ctrl$_{dhdt}$ experiment). In panel (c), the dashed light gray and blue lines represent the initial grounding-line positions for the Ctrl/damage experiments and the Ctrl$_{dhdt}$ experiment, and the black cross marks the location of the observed grounding-line position (Gardner et al., 2018).*

**22) L176-192 and Fig 3: So this part explains again my concern for Section 3.1. The RMSE of Ctrl_cal is even smaller than G1-G15. Does that mean we can calibrate the forcing data, e.g., basal melt rates, to get a better hindcast modeling result than tuning the damage parameters, or can we say that forcing is more important than damage?**

Response:

Thank you for your comment. The initial state of the Ctrl_cal (now Ctrl$_{dhdt}$) was explicitly adjusted to equilibrate toward a state that implicitly accounts for observed mass change rates. Given this, it is not surprising that it results in a smaller RMSE compared to the other experiments. However, such a simulation does not capture the driving mechanisms behind the mass change. In particular, it does not account for damage feedback mechanisms which, as we show, can have a strong influence on ice dynamics. Observations from satellite remote sensing highlight the widespread distribution and dynamic evolution of crevasses and rifts in the ice shelf and shear zones of Thwaites Glacier. This suggests that while the specific initialization procedure of Ctrl$_{dhdt}$ may improve the agreement with observations and hence better reproduce short-term mass loss, it does not necessarily remain valid on the long term given that it does not account for key processes which may have a significant influence on future ice loss.

As our results show, the long-term projections of Thwaites Glacier's evolution over 2020–2300 in the Ctrl$_{dhdt}$ experiment differ significantly from those produced when explicitly including ice damage (Figs. 6 and 7). Specifically, simulations accounting for ice damage predict more than twice the ice mass loss than the Ctrl$_{dhdt}$ experiment (Fig. 6).

**In Fig 3, I would also like to see the comparison of modeled and observed ice thickness data, which is a also very important information, or you might consider to add an additional figure for thickness.**

Response:

Thanks for your suggestion. We added panels (c), (g), and (k) showing the difference between the modeled and observed ice thickness in Figure 3 (now appears as Figure 4 in the revised manuscript):

[Figure]

***Figure 4.*** *Simulated ice velocity and ice thickness under different simulation experiments over the historical period 1990–2020. G1-16 denotes the simulation experiment in the ensemble of Group 1 ($C_1$=0.23, $C_{tr}$=0.26) that gives the most accurate (lowest RMSE) ice velocity simulation results. The $Ctrl_{dhdt}$ and Ctrl are the two simulation experiments of the model with deactivated damage processes (see Sect. 2 for details). (a), (e), and (i) show the spatial distribution of simulated ice velocity in the TG basin of different simulation experiments. (b), (f), and (j) show the difference between simulated and observed ice velocities. (c), (g), and (k) show the difference between simulated and observed ice thickness. (d), (h) and (l) show the comparison between simulated and observed ice velocities at each grid cell in the TG basin, with blue and red dots representing the grid cells of grounded ice and floating ice, respectively. In all maps, the dashed black lines are the observed grounding line (Gardner et al., 2018), the solid lines are the simulated grounding lines, and the light gray line is the basin boundary of the TG basin*

*derived from Zwally et al. (2015). The solid black curve in (a) is the central flowline profile stemming from the Antarctic surface flowline dataset developed by Liu et al. (2015)...."*

**23) Section 3.2: I'll hold my opinions for this section for now, as I do not see much information about SMB and basal melt forcings, e.g., if you use a coupling scheme between ice sheet and ocean model or you use a some parameterization approach. Before I have these information, I can't tell how meanful the projection numbers are in this section.**

Response:

All the simulations in Section 3.2 are continued from the historical state in 2020, i.e., under constant present-day climate conditions. We have thoroughly revised sections 2.2 and 2.3 and hope this clarifies our methodology.

As mentioned above, we would like to underline that this study does not aim to produce sea-level projections. Instead, we focus on testing the influence of ice damage on the Thwaites Glacier basin under constant present-day climate conditions. Our sensitivity experiments thus allow us to quantify TG's mass loss response to the damage feedback mechanism.

**24) Figure 6: you do have the current damage field. Then I think you probably want to compare them with some satellite images of crevasses. I think it is important for us to understand if the damage method you use is valid or not.**

Response:

Agreed. Please see our response to comment 14) above.

**Responses to comments from Reviewer Ravindra Duddu**

Damage intensity increases ice mass loss from Thwaites Glacier, Antarctica

We thank the editor and the two reviewers for their constructive feedback, which has helped us improve the manuscript. In response to the comments, we have thoroughly revised the manuscript. The key updates are the following:

- We have improved the writing, formulation and structure of the manuscript to enhance clarity.
- Appendix A has been removed and merged into section 2.1, where the model formulation has been extensively revised.
- To provide greater clarity on our methodology, we have introduced separate sections for the simulation protocol (Section 2.2) and the initialization (Section 2.3).
- Appendix B has been removed and integrated into the results section, which has been thoroughly revised to improve readability.
- A comparison of the modeled damage pattern and observations has been added (see Figure 5 in the revised manuscript).

Further details on these revisions are provided in our responses below. In the following, we use "**bold text**" for the reviewer's comments, "regular" text for our responses, and "*italic*" for text extracted from the manuscript.

**Comments to the Author**

**This article presents sensitivity studies using the Kori-ULB ice sheet model to explore the response of Thwaites Glacier (TG) with and without incorporating ice damage. Their main finding is that increasing damage intensity "results in larger retreat of the grounding line, higher ice velocity, thinner ice shelves, more ice mass loss, and a bigger contribution to global sea level rise." This is generally the consensus among researchers and is not in any way controversial nor groundbreaking; nevertheless, it is of important and interest to The Cryosphere community. The novelty of the article (in my opinion) lies in the set-up of the simulations for TG's with the calibration of the model using present day data for 1990, historical simulations over 1990-2020, and projections from 2020-2300. However, the description and explanation of the simulation set up and result can be improved.**
**The authors use the damage model proposed by Sun et al. (2012) that is based on the zero-stress theory. The contribution of this article is that it considers parametric studies by varying two damage parameters $C_1$ and $C_{tr}$, however, I had difficulty in following the model formulation and some of their equations seem to use inconsistent notation. The writing of this article is also not up to the standards of this journal and needs much improvement. The article must undergo**

**major revisions and re-review before it can be considered for publication.**

Response:

Thank you for your comments. The manuscript has been extensively revised by strengthening the description and explanation of the simulation set up and the results according the comments. For details, please see the following responses as well as the revised manuscript.

As suggested, we have revised the description of the model formulation in section "**2.1 Ice sheet and damage model**". To avoid redundancy, Appendix A has been removed. Further details are provided below in our point-by-point responses.

To address your feedback regarding the writing quality, we have thoroughly revised the manuscript to enhance its clarity, coherence, and overall readability.

**Detailed Comments:**

25) **In the abstract and elsewhere in the article, the authors use the term "strength of ice damage" I suggest replacing this term by "intensity of ice damage" so as not to confuse the reader with the strength of ice. Typically, a strength parameter is often used in ice damage models as a material property, so the terminology must be corrected.**

Response:

Thank you for your comment. We have replaced the term "strength of ice damage" by "intensity of ice damage" throughout the manuscript.

26) **The writing in the paper can be improved and may need professional writing help, as I found typos and grammatical errors. For example, in the abstract "ice sheet model enabled with the ice damage mechanics…" change as "ice sheet model including damage mechanics".**

Response:

We have revised the sentence from the abstract as you suggested.

(Line 12–14 in the revised manuscript without tracks): "…*Our results indicate that, when accounting for ice damage mechanics, the ice-sheet model captures the observed ice geometry and mass balance of Thwaites Glacier during the historical period (1990–2020)….*"

In addition, we have thoroughly revised the writing for greater clarity, coherence, and readability.

**27) The first sentence of the introduction begins with "Damage of glaciers …" For the sake of clarity and information, please define what you mean by damage in the context of ice sheet, especially at the scale you are defining damage. As we know, damage can be described at multiple scales all the way from microns (grain boundaries crack in the microstructure) to kilometers (rifts in ice shelves).**

Response:

Thank you for your comment. We consider large-scale damage of glaciers, resulting from the formation and development of both crevasses and rifts at the scale of meters to kilometers (hence no crack in the microstructure). We have revised the related sentence to make it clearer.

(Line 20–21): "…*The weakening of ice due to the formation of large-scale crevasses and rifts, known as damage, is gaining attention due to its impact on glacier and ice sheet evolution in a warming climate. …*"

**Also, the introduction is a bit sparse with references on damage models incorporated into either shallow shelf models or full Stokes models. As a suggestion, I would like to bring to the authors ice shelf scale studies incorporating damage into SSA (Huth et al., 2021b, 2023; Ranganathan et al., 2024).**

Response:

Thank you for your suggestion. We included these studies in our introduction section.

(Line 37–39): "…*Huth et al. (2021, 2023) integrated a creep damage model into a large-scale shallow-shelf ice flow model to simulate rift propagation leading to the formation of iceberg A68 from the Larsen C Ice Shelf.…*"

(Line 55–57): "…*Ranganathan et al. (2024) developed a damage evolution model coupled with a marine-terminating glacier flowline model and showed that damage can enhance mass loss from both grounded and floating ice.…*"

**28) Line 63 – Please clarify what you mean by 2.5D. Different authors use this in different ways. Is the thermal part 3D whereas the flow part 2D, is that right?**

Response:

That is correct. Kori-ULB is a plan-view vertically integrated model (and therefore two-dimensional) However, the temperature field is calculated in 3d in order to allow for a full thermomechanical coupling. We revised the related sentences.

(Line 81–83): "…*The Kori-ULB ice-sheet model (Pattyn, 2017; Coulon et al., 2024) is a vertically integrated, thermomechanical finite difference model that combines shallow-ice approximation with shallow-shelf approximation (so-called hybrid*

*model; Winkelmann et al., 2011)…."*

29) **On Line 70 – put CDM in parentheses, also CDM typically stands for continuum damage mechanics.**

Response:

Thank you for your comment. We revised the sentence as suggested.

**Also, change the next sentence as "Damage d(tau_1) includes a local source term d_1(tau_1) and an advection term due to ice flow d_tr." Also, damage is not a conserved variable. The damage evolution equation is not necessarily related to either mass or momentum but it rather a type of non-conserved phase field variable. Therefore, you should say "Damage advection due to ice flow …"**

Response:

We have revised those sentences as follows

(Line 100–102): "*…To determine the relationship between ice damage and the first principal stress $d(\tau_1)$, the CDM framework is based on two key components: a local source of damage term ($d_1$) that accounts for the local formation of damage, and an advection term ($d_{tr}$) that accounts for the transport of damage during ice flow. …*"

(Line 115–116): "*…In addition, damage fields are advected by ice flow. In this context, $d_{tr}$ represents the evolution of the vertically integrated damage field caused by advection, stretching, and mass loss or accumulation at the glacier's upper and lower surfaces. …*"

30) **I do not follow Eq. (3) for d(tau_1), you are taking max of certain quantities, but because there are so many parentheses, I am not able to follow what you are stating. Also, aren't d_s, d_s+d_b and C_1*h s all positive quantities so then why do you need to have the max(0,..) function, why can't you just simply take the max of the three non-zero quantities. I am also not clear how d_tr is defined in Eq. (4). Even though it may be well described in Sun et al., (2017) it would be useful discuss the definition of d_tr for completeness. The notation in the Appendix A is hard to follow as well (see my comments below on Appendix A).**

Response:

That is correct, $d_s$, $d_s + d_b$ and $C_1$ *h are all positive quantities, and damage is always positive. We have corrected Eq. (3) (now the Eq. (6) in the revised manuscript) as follows:

$$d_1(\tau_1) = min(d_s + d_b\,, C_1 * h)$$

$d_{tr}$ represents the damage transport during ice flow, which describes the evolution of the damage field due to advection, stretching, and the loss and accumulation of mass on the upper and lower surfaces of the glacier. For any time and position $(x, y, t)$, there is a field $d_l(x, y, t)$ calculated based on Eq. (3), and a field of transported crevasses depth $d_{tr}(x, y, t)$ which describes the transport of damage by ice flow. This transported crevasses depth $d_{tr}$ can be obtained by solving the damage transport equation (Sun et al., 2017):

$$\frac{\partial d_{tr}}{\partial t} + \nabla \cdot (\boldsymbol{u} d_{tr}) = -[max(\dot{a}, 0) + max(\dot{m}, 0)]\frac{d_{tr}}{h}$$

Where the left-hand side represents the vertically integrated damage conservation under ice flow, which includes the movement of the crevasses along with the ice flow and the stretching and compression. On the right-hand side, an increase in undamaged ice thickness is presumed to occur due to accumulation on the upper surface $(\dot{a})$, while the crevassed underside is eroded by basal melting $(\dot{m})$.

We have revised the description of the model formulation and added the definition of $d_{tr}$ in section "**2.1 Ice sheet and damage model**" as follows:

(Line 86–129): "*...In Kori-ULB, the relationship between the deviatoric stress $\tau$ and the strain rate $\dot{\epsilon}$ is described by Glen's constitutive flow law:*

$2A\tau^{n-1}\tau = \dot{\epsilon}$ ,                    *(1)*

*where A is Glen's flow law factor, dependent on the ice temperature, and n is the flow rate exponent, with n =3.*

*To investigate the dynamical response of the TG basin to ice damage and damage parametric perturbations, we couple the ice-sheet model with the continuum damage mechanics (CDM) model developed by Sun et al. (2017). This model establishes a direct link between the amount of damage and ice viscosity: the propagation of damage reduces the ice viscosity through Glen's flow law, leading to faster ice. This damage feedback is described by the integration of a damage factor $D(\tau)$ in Eq. (1):*

$2A\tau^2\tau = (1 - D(\tau))^3\dot{\epsilon}$ ,                    *(2)*

*with D (x, y, z) a scalar damage variable, taking values from 0 (undamaged ice) to 1 (ice entirely fractured by surface and basal crevasses). Given the integration over the vertical, this results in the following expression for the vertically integrated effective viscosity:*

$2h\mu = [h - d(\tau_1)]A^{-\frac{1}{3}}\dot{\epsilon}^{-\frac{2}{3}}$ ,                    *(3)*

*where $\mu$ is effective viscosity, h is ice thickness, d (x, y) $\in$ [0, h (x, y)] is the vertical integral of D (x, y, z), and $\tau_1$ is the first principal stress. To determine the relationship between ice damage and the first principal stress $d(\tau_1)$, the CDM framework is based on two key components: a local source of damage term $(d_l)$*

that accounts for the local formation of damage, and an advection term ($d_{tr}$) that accounts for the transport of damage during ice flow.

In the absence of advection, ice damage is expressed as the total depth of the crevasses, i.e., the sum of surface crevasses $d_s$ and basal crevasses $d_b$ (Nick et al., 2011, 2013; Cook et al., 2014; Sun et al., 2017). Those can be calculated by the zero-stress assumption (Nye, 1957; Nick et al., 2011):

$$d_s = \frac{\tau_1}{\rho_i g} + \frac{\rho_w}{\rho_i} d_w, \tag{4}$$

$$d_b = \frac{\rho_i}{\rho_w - \rho_i} \left( \frac{\tau_1}{\rho_i g} - H_{ab} \right), \tag{5}$$

where $d_w$ is the water depth in the surface crevasse (here we only consider dry crevasses, so $d_w$ is equal to 0), $H_{ab}$ is the thickness above floatation, $g = 9.81 \ m \ s^{-2}$ is the gravitational acceleration, and $\rho_i = 917 \ kg \ m^{-3}$ and $\rho_w = 1028 \ kg \ m^{-3}$ are the ice and seawater density, respectively.

The local source of damage term $d_1(\tau_1)$ is then expressed as

$$d_1(\tau_1) = min(d_s + d_b, C_1 * h), \tag{6}$$

where $C_1$ is a parameter ranging from 0 to 1 that sets an upper limit to $d_1(\tau_1)$ as a fraction of the ice thickness. This constraint prevents an overestimation of crevasse depth in the gridded domain.

In addition, damage fields are advected by ice flow. In this context, $d_{tr}$ represents the evolution of the vertically integrated damage field caused by advection, stretching, and mass loss or accumulation at the glacier's upper and lower surfaces. The transported crevasses depth $d_{tr}$ can be solved by the following damage transport equation (Sun et al., 2017):

$$\frac{\partial d_{tr}}{\partial t} + \nabla \cdot (\boldsymbol{u} d_{tr}) = -[max\,(\dot{a}, 0) + max\,(\dot{m}, 0)\,] \frac{d_{tr}}{h}, \tag{7}$$

The left-hand side of Eq. (7) represents the conservation of vertically integrated damage, which includes the advection of crevasses with the ice flow and the effect of stretching and compression. On the right-hand side, damage reduction is modeled through two processes: an increase in undamaged ice thickness due to surface accumulation ($\dot{a}$) and erosion of the crevassed ice bottom by basal melting ($\dot{m}$).

Overall, at any given time and position (x, y, t), there exist two damage fields: the locally generated crevasse depth $d_1$ (x, y, t), as calculated above, and the advected crevasses depth $d_{tr}$ (x, y, t). Assuming that crevasse surfaces do not bond together during closure, at least on the timescale relevant to crevasse closure (Sun et al., 2017), the final expression of damage d (x, y, t) is given by

$$d(x, y, t) = min(C_{tr} * h(x, y, t), max\,(d_1(x, y, t), d_{tr}(x, y, t))), \tag{8}$$

*where $C_{tr}$ is a parameter that limits d as a fraction of the ice thickness, with $C_1 \leq C_{tr}$. This implies that regions of the ice shelf subjected to lower stress inherit damage from the upstream areas that are experiencing higher stress...."*

31) **Line 90 – Was the inversion to obtain ice sheet initial conditions only performed once and was used as the starting point for all simulations. Also, more details on the inverse would be useful for the general reader, unless Coulon et al. (2024) has it all in detail. Please add a sentence to clarify.**

Response:

The nudging scheme described here is used to obtain the ice sheet initial conditions used as a starting point for both the Ctrl experiment as well as the 43-member ensemble used to quantify TG's sensitivity to the intensity of ice damage. In contrast, the initial state used for the Ctrlcal experiment was produced using the method described in van den Akker et al. (2025). Note that the only difference between both methods is that in the second case, the satellite-based data of present-day ice mass-change rates (dhdt) is added to the surface mass balance used as a boundary condition in the initialization. To avoid confusion, we have replaced 'Ctrl$_{cal}$' with 'Ctrl$_{dhdt}$' throughout the entire manuscript.

The initialization procedure is described in detail in **Appendix A** of Coulon et al. (2024). In the revised manuscript, we have improved the description of the initialization procedure in **section 2.3**:

(Line 154–170): "*...The initial conditions for both the 43-member damage ensemble and the Ctrl experiment are obtained by an inverse simulation nudging towards present-day ice-sheet geometry (Pollard and DeConto, 2012; Bernales et al., 2017; Coulon et al., 2024), using present-day ice-sheet surface and bed geometry from BedMachine v2 (Morlighem et al., 2020) and present-day surface mass balance and air temperature from the polar regional climate model MARv3.11 (Kittel et al., 2021). A detailed description of the initialization procedure is provided in Appendix A of Coulon et al. (2024). The initial state for the Ctrl experiment is identical to that of the 43-member damage ensemble, ensuring that all start from the same ice sheet geometry. In the damage sensitivity experiments, ice damage is activated from the first timestep of the historical simulation, meaning that the ice sheet is considered undamaged at the start of 1990. Given that this assumption is somewhat idealized, the simulated damage can be interpreted as relative to the initial state.*

*To reproduce the dynamic disequilibrium observed during the historical period, we apply the initialization method of van den Akker et al. (2025). Specifically, the initial state of the Ctrl$_{dhdt}$ experiment is obtained by adding a 'correction term' – equal to minus the observed mass change rates (taken from Bevan et al., 2023) – to the present-day surface mass balance (Kittel et al., 2021) during the transient nudging procedure. This ensures that, by the time the nudging procedure has*

*achieved a steady geometry, the model has been trained to produce ice fluxes that closely match observations. In other words, the ice sheet model is 'trained' to equilibrate toward a state that implicitly accounts for observed mass change rates. As a result, it is important to note that the Ctrl$_{dhdt}$ experiment starts from a slightly different initial state than the Ctrl and damage experiments (see supplementary Figs. S1 and S2)....*"

Coulon, V., Klose, A. K., Kittel, C., Edwards, T., Turner, F., Winkelmann, R., and Pattyn, F.: Disentangling the drivers of future Antarctic ice loss with a historically calibrated ice-sheet model, The Cryosphere, 18, 653–681, doi:10.5194/tc-18-653-2024, 2024.

van den Akker, T., Lipscomb, W. H., Leguy, G. R., Bernales, J., Berends, C., van de Berg, W. J., and van de Wal, R. S. W.: Present-day mass loss rates are a precursor for West Antarctic Ice Sheet collapse, The Cryosphere, 19, 283–301, doi:10.5194/tc-19-283-2025, 2025.

**32) Line 92 – Assuming that present day is undamaged is unrealistic, but it is an assumption. Perhaps, you can add a clarification that it can be interpreted as relative damage with respect to the initialized state. In the sentence below you report RMSE, perhaps it is a bit more helpful to report relative RMSE as a percentage.**

Response:

Thank you for your comment. We have added the following clarification

(Line 159–162): "*.... In the damage sensitivity experiments, ice damage is activated from the first timestep of the historical simulation, meaning that the ice sheet is considered undamaged at the start of 1990. Given that this assumption is somewhat idealized, the simulated damage can be interpreted as relative to the initial state....*"

In addition, we have followed your suggestion and now use the relative RMSE. We have added the equations of RMSE and the relative RMSE and revised the related description in **section 2.3**.

(Line 171–178): "*...To evaluate the modeled initial conditions, we compute the root mean square errors (RMSE) and the relative RMSE (rRMSE) between simulated and observed ice velocity (Rignot et al., 2017) and ice thickness (Morlighem et al., 2020):*

$$RMSE = \sqrt{\frac{\sum_{i=1}^{n}(Sim_i - obs_i)^2}{n}} \qquad (9)$$

$$rRMSE = \frac{RMSE}{\overline{obs}}, \qquad (10)$$

*where n is the number of grid points, sim$_i$ and obs$_i$ are the simulated and observed ice velocity (Rignot et al., 2017) or thickness (Morlighem et al., 2020), respectively,*

*and $\overline{obs}$ is the mean observed ice velocity or thickness. In addition, we estimate the mean distance between the modeled and observed grounding-line position using the "open-ended box" approach of Moon and Joughin (2008)...."*

(Line 179–185): *"...Following the standard initialization procedure (used in the Ctrl and damage experiments), the RMSE (rRMSE) values between simulated and observed ice velocity and thickness are 201 m $a^{-1}$ (1.66) and 28 m (0.01) for the whole basin, and 786 m $a^{-1}$ (0.98) and 28 m (0.1) for floating ice only (supplementary Fig. S1). The modeled grounding-line position of the TG basin is in good agreement with observations (Gardner et al., 2018), with an average offset of 1.3 km. For the initial state of the $Ctrl_{dhdt}$ experiment, the RMSE (rRMSE) values of ice velocity and thickness are 172 m $a^{-1}$ (1.42) and 27 m (0.01) for the whole basin, and 659 m $a^{-1}$ (0.83) and 54 m (0.13) for floating ice only (supplementary Fig. S2). The modeled grounding-line position also closely aligns with observations, with an average offset of 2.3 km...."*

**33) Line 98 – The term local damage is used to define the damage production term to highlight the fact that damage can also advect from upstream. However, both advection and production terms are local damage whereas nonlocal damage refers to those approaches that incorporate a nonlocal length scale (Duddu, 2020; Huth 2021b).**

Response:

Thank you for your comment. We agree with your description. The advection and production terms described in our study are local damage. To avoid confusion, we have revised this section as follows.

(Line 100–102): *"...To determine the relationship between ice damage and the first principal stress $d(\tau_1)$, the CDM framework is based on two key components: a local source of damage term $(d_1)$ that accounts for the local formation of damage, and an advection term $(d_{tr})$ that accounts for the transport of damage during ice flow. ..."*

We have also deleted the sentence "$C_l$ sets a limit on local damage and $C_{tr}$ sets a limit on total damage".

**Also, on Line 101, why not say parameter values, why use the term parameter members. I would replace the word members with values in this context throughout the paper.**

Response:

We have replaced "members" by "values" as you suggested throughout the paper.

**34) Line 114 – Add statement to clarify what "different physics" the Ctrl_cal**

**experiments consider as opposed to the damage experiments.**

Response:

Apologies for the confusion. The physics are the same, it is the initialization procedure which is different. To derive the initial state for the Ctrl$_{cal}$ (now the Ctrl$_{dhdt}$) experiment, the satellite-based data of present-day ice mass-change rates (dhdt) is added to the surface mass balance used as a boundary condition during the initialization. To avoid confusion, we have replaced 'Ctrl$_{cal}$' with 'Ctrl$_{dhdt}$' throughout the entire manuscript.

To enhance clarity, we have removed this sentence and revised the related sentences.

(Line 149–150): "...*In addition, two control simulations without damage serve as baselines for comparison: one designed to reproduce observed mass-change rates (Ctrl$_{dhdt}$), and another without this constraint (Ctrl)....*"

(Line 158–161): "...*The initial state for the Ctrl experiment is identical to that of the 43-member damage ensemble, ensuring that all start from the same ice sheet geometry. In the damage sensitivity experiments, ice damage is activated from the first timestep of the historical simulation, meaning that the ice sheet is considered undamaged at the start of 1990....*"

(Line 168–170): "...*As a result, it is important to note that the Ctrl$_{dhdt}$ experiment starts from a slightly different initial state than the Ctrl and damage experiments (see supplementary Figs. S1 and S2)....*"

For additional details regarding the initialization procedures for both the Ctrl$_{dhdt}$ experiment and the Ctrl/damage experiments, please refer to our response to **Comment 7)** as well as **Section 2.3** of the revised manuscript.

35) **Line 141 – Replace the term "the strength of ice damage" with "the intensity of ice damage" throughout the paper. In Table 1, the acronym SLC is not**

Response:

Thanks for your suggestion, we have replaced the term "the strength of ice damage" with "the intensity of ice damage" throughout the paper.

SLC represents the contribution to sea level. We have revised the relevant sentence to clarify this when the full name of SLC appears early in **Section 2.2**.

36) **Line 150 – The description here could be improved. Is it correct that so this ignoring damage underestimates ice mass change by more than an order of magnitude, that is 2.1 (without damage) instead of 38.3 (with damage) or 28.1 (Ctrl_cal)?**

Response:

Thanks for your suggestion. This is correct. We have modified the description of the results as follows

(Line 206–211): "…*For the period 1990–2020, the simulated mean net mass balance for Group 1 (with damage) is -26.5 Gt a$^{-1}$, which is comparable to satellite-derived observations (-46.1 ± 7.2 Gt a$^{-1}$ over 1992–2017; mean ± 1 s.d.). In contrast, neglecting ice damage underestimates ice mass change by more than an order of magnitude, with the Ctrl experiment simulating only 1.2 Gt a$^{-1}$. The Ctrl$_{dhdt}$ experiment, which also ignores ice damage but applies an artificial correction to the ice mass-change rate, yields a simulated net mass balance of -30.1 Gt a$^{-1}$ over 1990–2020, comparable to estimates from Group 1.…*"

37) **Figure 4 – This is an important figure, and the results can be explained better with a sentence. The way I understand the light red and green corresponding to the lowest damage do not produce any significant retreat compared to the observed GL and central profile. Is the black line in subfigure (b) the observed elevation, please clarify.**

Response:

That is correct. Along the central profile, the light red and green corresponding to the lowest damage do not produce significant retreat compared to the observed GL. However, the two lowest damage scenarios still result in an obvious retreat throughout the entire TG basin, such as in regions in the upstream glacier area of the TEIS, compared to the observed GL. For clarity, the figure previously referred to as Figure 4 appears as Figure 7 in this revised submission. We added the following sentence into the revised manuscript:

(Line 327–330): "…*Ensemble members with lower damage intensities (light red and green lines in Figs. 7a and 7b) also show less retreat compared to the initial grounding line position. However, even these cases exhibit noticeable retreat across the TG basin, particularly in the upstream glacier area of TEIS.…*"

The black line in subfigure (b) indeed represents the observed elevation. We have now removed it from Figure 7 (revised) and substituted it with the initial elevation for comparative purposes.

38) **Line 219 – The sea level rise by year 2300 is about 5 – 8 cm. This seems quite small. Is this cumulative sea lever rise or is it sea level rise. From what I recall, the question always was if Antarctica could contribute to a meter or more of sea level rise by 2300. Does this mean that Thwaites is not going to be a major contributor for sea level rise?**

Response:
Thank you for your comment. In our simulations, the boundaries of the basin

remain fixed, preventing us from directly inferring global mean sea-level rise. It is important to note that the goal of this study is not to produce sea-level projections. Instead, we focus on testing the influence of damage on the Thwaites Glacier basin under constant present-day climate conditions. Our sensitivity experiments quantify TG's mass loss response to the damage feedback mechanism. Our results show that damage accelerates grounding-line retreat and ice mass loss in the TG basin, suggesting that it is a key process that should be considered when modeling TG's future evolution.

We have revised the text to refer to mass loss in sea-level equivalent rather than as a direct sea-level contribution.

(Line 317–318): "...*By 2300, the simulated mean ice mass loss in Group 1 reaches 5.5 ± 3.3 cm sea-level equivalent – 5 times higher than in the Ctrl simulations (1 cm) and more than twice that of the $Ctrl_{dhdt}$ (2 cm) experiments (Fig. 6a)....*"

39) **Line 233 – It is stated that "The increased ice velocity and decreased ice thickness further stimulate damage formation and propagation." I am not sure if decreased ice thickness would always stimulate damage formulation. As the ice thickness is reduced the driving stress could also be reduced and this could reduce damage formation. Perhaps, my reasoning is wrong, but the authors could comment on this.**

Response:

Thank you for your comments. Indeed, the driving stress is linear to the ice thickness, so reducing the ice thickness could reduce damage. Nonetheless, our damage primarily depends on strain rates, which increase faster than the reduction in ice thickness. As a result, the overall effect is an increase in the formation and propagation of damage. We have revised the relevant sentences accordingly.

(Line 294–298): "...*Ice-shelf thinning and weakening, reproduced in both experiments over the historical period (Figs. 4c and 4g), lead to increased ice velocity and decreased upstream ice thickness. In G1 simulations, this further stimulates damage formation and propagation, amplifying mass loss. Although reduced ice thickness could decrease the driving stress, potentially limiting damage formation, our model damage primarily depends on strain rates, which increase in thinning ice shelves as buttressing is reduced....*"

40) **Line 265 – The long-term projections of TG's evolution (2020-2300) differ between the Ctrl_cal and those with damage. It would be good to remind the reader, quantitatively and qualitatively what are these differences by 2300. Also, important to note that we will not know which of the two projections is realistic. I do not think we can simply state that incorporating damage mechanics improves the projections without some validation. This needs more**

**discussion.**

Response:

Thank you for your relevant suggestion. We have quantitatively and qualitatively described the differences in mass loss, ice velocity and grounding line retreat by 2300 between the Ctrl$_{dhdt}$ experiment and those with damage by adding the following paragraph:

(Line 385–388): "*...However, mass loss projections for 2020–2300 in the Ctrl$_{dhdt}$ experiment diverge significantly from those that explicitly represent ice damage processes. Specifically, simulations accounting for ice damage predict more than twice the ice mass loss, higher mean ice velocity along the central flowline profile, and greater inland retreat of the grounding line (Figs. 6 and 7)....*"

It is true that we cannot say which of the two projections is realistic. While accounting for damage allows us to capture the observed mass loss over the historical period, we also show that the positive feedback between damage processes and ice-shelf weakening leads to a substantial increase in mass loss on multi-decadal to centennial scales. This increase is particularly pronounced when compared to the Ctrl$_{dhdt}$ experiment, in which the observed trend in mass loss has simply been extended into the future.

The application of prognostic modeling to investigate the impact of damage on the dynamic evolution of ice shelves, as performed in this study, remains in its early stages. Further work is needed to refine how damage is represented in ice-sheet models, particularly through improved validation against observations. Nevertheless, our results suggest that damage is a key mechanism accelerating grounding-line retreat, with the associated ice mass loss representing a potential contribution to future sea-level rise. We have added a statement along those lines in the discussion section.

**41) Line 330 – Grant number is missing and typed as XX**

Response:

We added the grant number and revised the *Acknowledgements* section.

(Line 454–461): "*...Acknowledgements. YL and QY received support from the National Natural Science Fund of China (No. 42406242), the Southern Marine Science and Engineering Guangdong Laboratory (Zhuhai) (Nos. SL2021SP201 and SML2022SP401), and the Fundamental Research Funds for the Central Universities, Sun Yat-sen University (No.74110-31610046). This research was also supported by OCEAN:ICE (which is co-funded by the European Union, Horizon Europe Funding Programme for research and innovation under grant agreement Nr. 101060452 and by UK Research and Innovation) and the HiRISE (NWP GROOT, Netherlands, under grant agreement No. OCENW.GROOT.2019.091).*

*Computational resources have been provided by the Consortium des Équipements de Calcul Intensif (CÉCI), funded by the Fonds de la Recherche Scientifique de Belgique (F.R.S.-FNRS) under Grant No. 2.5020.11 and by the Walloon Region....."*

**42) Appendix A – This section needs to undergo a thorough revision. Please see the comments below:**

Response:

Appendix A has been removed and merged into section 2.1., where the model formulation has been extensively revised.

- **The notation is poorly chosen, for example stress and strain are tensors, but they are denoted as scalars. Eqs. (A1) and (A2) need to be corrected.**

Response:

We corrected the equations as you suggested.

- **In Eq. (A3) the first term on the RHS is (h-tau_1), which does not make sense because h ice thickness and tau_1 is the first principal stress. You cannot subtract two quantities with different units.**

Response:

Thank you for noticing this. Indeed, something was wrong with the expression of Eq. (A3). We have revised it.

$$2h\mu = [h - d(\tau_1)]A^{-\frac{1}{3}}\dot{\epsilon}^{-\frac{2}{3}}$$

- **Line 359 tau_1 cannot be determined by setting it equal to depth of crevasses, stress and depth are different physical quantities.**

Response:

Agreed. We revised this section as follows:

(Line 103–104): "...*In the absence of advection, ice damage is expressed as the total depth of the crevasses, i.e., the sum of surface crevasses $d_s$ and basal crevasses $d_b$ (Nick et al., 2011, 2013; Cook et al., 2014; Sun et al., 2017)....."*

- **In Eq. (A8) u should be bold as it represents the ice velocity, which is a vector, otherwise the divergence operator does make any sense.**

Response:

We have revised the equation as you suggested.

(Line 118): "...

$$\frac{\partial d_{tr}}{\partial t} + \nabla \cdot (\boldsymbol{u} d_{tr}) = -[max\,(\dot{a}, 0) \; + max\,(\dot{m}, 0)\,]\frac{d_{tr}}{h}$$

*...''*

- **I do not understand how the sentence above Eq. (A9) about crevasse closure leads to the specific definition of damage in (A9)**

Response:

Thank you for your comment. We adopt the assumption proposed by Sun et al. (2017) that crevasses surfaces do not bond together during the closure process, at least within the timescale of crevasse closure. This assumption implies that regions of the ice shelf under lower stress inherit damage from any higher-stress region upstream, i.e. that $d(\tau_1) = max\,(d_1(\tau_1), d_{tr})$.

Note that we additionally apply a constraint based on a fraction of ice thickness ($C_{tr}$ * h) to set an upper limit on the final damage field, and integrate these three components to define the final damage field.

We have revised the formulation of the description of the model in section "**2.1 Ice sheet and damage model**" part accordingly. For details, please see the response to **comment 6)** as well as the revised manuscript.

- **Line 379 – Please explain what the term "model collapse before 2300" means.**

Response:

Since the boundaries of the drainage basin of Thwaites Glacier remain fixed, the model run encounters numerical instability and eventually stops whenever the grounding line approaches those boundaries. This numerical failure is therefore linked to the collapse of the basin itself, meaning that the instability arises from both numerical and physical causes. We have modified this section as follows:

(Line 331–334): "*...In Group 2 (high damage intensity members, Table 1), 18 out of 27 simulations resulted in model failure before 2300 (dashed dark red lines in Fig. 8a). Since the Thwaites Glacier drainage basin boundaries remain fixed, the model encounters numerical instability and eventually stops when the grounding line approaches these boundaries. This numerical failure is thus linked to basin collapse and arises from both numerical and physical instabilities....*"

- **Line 381 – it is not clear what is meant by "averagely" in the next sentence, maybe say "on an average" instead. Also, 7 – 10 cm of global sea level rise seems on the lower end, when other works are exploring the possibility of 1 – 3 m of sea level rise (e.g. DeConto and Pollard, 2016).**

Response:

Thank you for your suggestion. We have revised the text as you suggested.

(Line 336–338): "*...In comparison, the 18 members from G2$_{ext}$ show an average ice*

*mass loss of 7.1 ± 2.8 cm by 2100 (dark red line and hatched area in Fig. 8b), ...”*

The relatively limited contribution of ice mass loss to sea level from the TG basin simulated in this study may be due to the use of present-day atmospheric and oceanic forcing in our forward simulations rather than projections based on future climate scenarios. This constraint likely results in a lower SLC over the 280-year simulation period (2020–2300), compared to the projection results of DeConto and Pollard (2016) that employed high-resolution RCM-based atmospheric forcing under three extended Representative Concentration Pathway (RCP) scenarios.

Additionally, note that the boundaries of the basin cannot move in our simulations, which does not allow us to make a robust inference on global mean sea-level rise. As mentioned previously, this study does not aim to produce sea-level projections but rather assess the impact of damage on projected mass loss.

- **Table A1 – How is RMSE calculated, please give more details and perhaps consider relative RMSE to report it as percentages.**

Response:

The RMSE was calculated as follows:

$$RMSE = \sqrt{\frac{\sum_{i=1}^{n}(Sim_i - obs_i)^2}{n}}$$

where $n$ is the number of grid points. $sim_i$ and $obs_i$ are the simulated and observed ice velocity (Rignot et al., 2017) or thickness (Morlighem et al., 2020), respectively.

We have also added the relative RMSE as you suggested. The relative RMSE (rRMSE) between simulated and observed ice velocity and ice thickness is calculated by:

$$RMSE = \frac{RMSE}{\overline{obs}}$$

where $\overline{obs}$ is the mean observed ice velocity or thickness.

We have added the relative RMSE into Table A1 and also revised Table A1 as follows (Please note that Table A1 has now been moved to the supplementary material as Table S1):

(In Supplementary): "...

***Table S1***. *Summary of the damage sensitivity experiments and two control experiments performed at the TG basin under constant present-day conditions. The values of parameters $C_1$ and $C_{tr}$ of the 43 simulations considering the damage processes are produced using Latin hypercube sampling in their parameter space. The RMSE and rRMSE between the simulated and observed ice velocity (Rignot et al., 2017) are used to evaluate the accuracy of the model results. The experiments*

*marked with an asterisk\* in Group 2 correspond to 18 simulations that predominantly exhibit high damage intensity, leading to model failure before 2300; these simulations are collectively referred to as $G2_{ext}$ in this study.*

| Scenarios | ID | Damage parameters | | RMSEs (rRMSEs) over 1990–2020 | | |
| | | $C_1$ | $C_{tr}$ | RMSE (rRMSE) (whole basin) | RMSE (rRMSE) (floating ice) | RMSE (rRMSE) (grounded ice) |
|---|---|---|---|---|---|---|
| Ctrl

deactivated damage processes | | – | – | 190.9 (1.58) | 745.2 (0.92) | 95.6 (1.15) |
| $Ctrl_{dhdt}$

deactivated damage processes;
corrected SMB using satellite-observed ice mass-change rates (Bevan et al., 2023) | | – | – | 181.3 (1.5) | 752.7 (0.97) | 71 (0.84) |
| Group 1
damage processes;

the contribution to Sea-level (SLC) within the range of observational estimates ± 2 s.d. (0.24 ± 0.08 cm over 1992–2017) in the historical simulation (Shepherd et al., 2019) | 1 | 0.0308 | 0.9861 | 606.6 (5.03) | 2468.7 (3.12) | 77.8 (1.04) |
| | 2 | 0.0576 | 0.7712 | 327.7 (2.72) | 1394.3 (1.68) | 67.1 (0.87) |
| | 3 | 0.0585 | 0.5215 | 221.1 (1.84) | 928.1 (1.12) | 62.5 (0.8) |
| | 4 | 0.0657 | 0.3400 | 196.8 (1.63) | 815 (0.98) | 63.6 (0.81) |
| | 5 | 0.0806 | 0.6590 | 271.4 (2.25) | 1137.4 (1.37) | 65.4 (0.85) |
| | 6 | 0.0838 | 0.5067 | 233 (1.93) | 974.2 (1.16) | 63.4 (0.82) |
| | 7 | 0.0909 | 0.3521 | 194.8 (1.62) | 805.4 (0.97) | 60.4 (0.77) |
| | 8 | 0.0951 | 0.5530 | 249.4 (2.07) | 1041 (1.25) | 63.9 (0.83) |
| | 9 | 0.1014 | 0.3265 | 192.3 (1.6) | 794.2 (0.95) | 60.2 (0.77) |
| | 10 | 0.1297 | 0.3007 | 192.3 (1.6) | 791.1 (0.95) | 59.2 (0.76) |
| | 11 | 0.1385 | 0.4258 | 234.3 (1.95) | 967.1 (1.17) | 62.8 (0.82) |
| | 12 | 0.1399 | 0.2846 | 189.4 (1.57) | 776.2 (0.93) | 59.7 (0.77) |
| | 13 | 0.1780 | 0.2613 | 185.9 (1.54) | 760 (0.91) | 60.2 (0.77) |
| | 14 | 0.1819 | 0.2600 | 185.6 (1.54) | 757.8 (0.91) | 60.3 (0.77) |
| | 15 | 0.1932 | 0.2591 | 185.2 (1.54) | 756.2 (0.91) | 60.2 (0.77) |
| | 16 | 0.2255 | 0.2588 | 184.9 (1.54) | 754.4 (0.91) | 59.9 (0.77) |
| Group 2:
damage processes;
SLC outside the range of observational estimates ± 2 s.d. in | 1 | 0.0174 | 0.9257 | 178.8 (1.48) | 709.7 (0.88) | 77.6 (0.95) |
| | 2 | 0.0249 | 0.1666 | 181.1 (1.5) | 705.8 (0.87) | 87.3 (1.06) |
| | 3 | 0.0429 | 0.3302 | 193.2 (1.6) | 795.9 (0.96) | 67.3 (0.85) |
| | 4* | 0.0682 | 0.9409 | 560.8 (4.65) | 2217 (2.76) | 106.7 (1.48) |
| | 5 | 0.0778 | 0.1783 | 184.1 (1.53) | 755.1 (0.91) | 64.8 (0.82) |

| | | | | | |
|---|---|---|---|---|---|
| the historical | 6* | 0.1232 | 0.9702 | 1092 (9.09) | 3962.9 (5.05) | 260.8 (3.95) |
| simulation | 7* | 0.1381 | 0.8655 | 554.8 (4.59) | 2134.1 (2.66) | 128.5 (1.81) |
| | 8* | 0.1486 | 0.6384 | 489.8 (4.06) | 1980.6 (2.39) | 104.6 (1.44) |
| | 9 | 0.1579 | 0.5134 | 270 (2.24) | 1114.8 (1.35) | 63 (0.83) |
| | 10* | 0.1759 | 0.8237 | 427.3 (3.52) | 1603 (1.99) | 126.6 (1.79) |
| | 11 | 0.1807 | 0.4473 | 253.3 (2.1) | 1043.6 (1.26) | 63.1 (0.83) |
| | 12* | 0.1911 | 0.6941 | 404.2 (3.34) | 1586.3 (1.94) | 102.6 (1.42) |
| | 13* | 0.2267 | 0.7377 | 403 (3.32) | 1508.7 (1.87) | 123.9 (1.75) |
| | 14 | 0.2335 | 0.3962 | 246.7 (2.05) | 1015 (1.23) | 62.9 (0.83) |
| | 15 | 0.2411 | 0.4244 | 301.7 (2.5) | 1246.1 (1.5) | 67.3 (0.9) |
| | 16* | 0.2604 | 0.467 | 512.1 (4.25) | 2076.9 (2.48) | 112.1 (1.55) |
| | 17 | 0.2812 | 0.3218 | 212.7 (1.77) | 868.6 (1.05) | 62 (0.81) |
| | 18* | 0.3068 | 0.648 | 359.8 (2.97) | 1334.9 (1.64) | 124.1 (1.76) |
| | 19* | 0.3497 | 0.6175 | 552.9 (4.56) | 2057.9 (2.52) | 165.6 (2.39) |
| | 20* | 0.3674 | 0.6608 | 1186.3 (9.73) | 4249.6 (5.27) | 301.6 (4.61) |
| | 21* | 0.3789 | 0.4358 | 526.8 (4.37) | 2140.1 (2.53) | 118.2 (1.63) |
| | 22* | 0.3877 | 0.6057 | 622.4 (5.13) | 2291 (2.8) | 190.8 (2.79) |
| | 23* | 0.4129 | 0.5769 | 574 (4.74) | 2140.6 (2.57) | 183.5 (2.66) |
| | 24* | 0.428 | 0.5242 | 273.4 (2.27) | 1018.8 (1.22) | 109.9 (1.53) |
| | 25* | 0.4538 | 0.5433 | 289.8 (2.4) | 1075.1 (1.29) | 112.1 (1.58) |
| | 26* | 0.4711 | 0.5318 | 262 (2.17) | 983.3 (1.18) | 100.3 (1.4) |
| | 27* | 0.5202 | 0.5657 | 624 (5.15) | 2330.4 (2.81) | 192.7 (2.79) |

*..."*

We have also added the description of relative RMSE into the revised manuscript:

(Line 248–252): "*...By 2020, the mean RMSEs (rRMSEs) of simulated ice velocities in Group 1 and Group 2 simulations are 241 ± 102 m a⁻¹ (1.6) and 435 ± 247 m a⁻¹ (1.9) for the whole basin, and 995 ± 417 m a⁻¹ (1.2) and 1665 ± 880 m a⁻¹ (1.5) for floating ice only. For comparison, the RMSEs (rRMSEs) between observed and simulated ice velocities in the Ctrl$_{dhdt}$ and Ctrl experiments are 181 m a⁻¹ (1.5) and 191 m a⁻¹ (1.58) for the whole basin, and 753 m a⁻¹ (0.97) and 745 m a⁻¹ (0.92) for floating ice only. ...*"

- **Figure A1 – why is the flowline marked in subfigures (a) and (b)**

Response:

Thank you for noticing. We have removed the flowline from subfigures (a) and (b) in the original Figures A1 and A2, which now appear as Supplementary Figures S1 and S2 respectively.

- **Figure A3 – what is the reason the gray and blue model lines are not matching**

**well in the top region but matching well in the bottom region, which is apparent from subfigure (b).**

Response:

This difference likely stems from the distinct initial states used in the Ctrl (gray line) and the Ctrl$_{dhdt}$ (blue line) experiments. The initial state of the Ctrl$_{dhdt}$ experiment is generated by adding a 'correction term' equal to minus the observed mass change rates to the present-day surface mass balance during the transient nudging procedure, following the method described in van den Akker et al. (2025). This results in a different initial state compared to the Ctrl experiment. After 30 years of historical simulation, this difference leads to variations in the evolution of the grounding line in the TG basin. Our results (now appears as Figure 3 in the revised manuscript) indicate that in the upper region, the Ctrl$_{dhdt}$ experiment shows more thinning, ice flow acceleration, and grounding-line retreat compared to the Ctrl experiment. In contrast, in the lower region, a generally more stable area, differences between the two control experiments remain relatively minor.

- **Figure A4 – please add a sentence to clarify how ice mass loss is calculated.**

Response:

We have taken the decision to remove this figure from our revised manuscript.

- **Figure A5 – what does model collapse mean.**

Response:

See response to the comment about **Line 379 above.**

---

## Author Response (AR2)

**egusphere-2024-2916**

**Responses to comments from Reviewer Ravindra Duddu**

Damage intensity increases ice mass loss from Thwaites Glacier, Antarctica

We thank the editor and the reviewer for their constructive feedback and continued support, which has helped us improve the manuscript. In response to the comments, we have thoroughly revised the manuscript. Further details are provided in our responses below. In the following, we use "**bold text**" for the reviewer's comments, "regular" text for our responses, and "*italic*" for text extracted from the manuscript.

**Comments to the Author**

**This is the second time I am reviewing this article. The authors have adequately responded to both reviewer comments and extensively revised the article. The figures and the text in the article are much clearer, but there are still some notation inconsistencies and a few statements that need revision. As this is first of a kind study exploring the effect of damage on ice flow of a real (Thwaites) glacier, I recommend it for publication after the following comments are addressed. However, there are discrepancies between observations and simulation results, lack of understanding of how represent basal crevasse formation, and the damage model has a few deficiencies (due to the zero-stress approximation), which requires further research. This could be elaborated in the discussion and conclusion sections better.**

Response:

> We sincerely thank you for your thoughtful suggestions and constructive comments, which have been highly valuable in improving the quality of this manuscript. We also appreciate your thorough re-evaluation of the revised manuscript. We have carefully revised the manuscript based on your suggestions and comments. Please find our detailed responses to each of your specific comments below.

**Detailed Comments:**

1) **The following two sentences in the Introduction near Line #45 need to be revised.**
   **- "They have one critical limitation, i.e., being diagnostic, which means that they investigate the instantaneous effect of damage on ice dynamics, but not the evolution of damage when ice thickness is allowed to evolve according to the applied changes."**
   **In the references cited in this paragraph, Huth et al. (2021, 2023) and Duddu et al (2020) do consider ice thickness changes. Huth et al solve the shallow shelf equations including the ice thickness evolution and Duddu et al. use an updated**

**Lagrangian implementation to account for ice thickness evolution. However, because these works considered smaller time scales (months to years) there is no significant effect of damage on ice flow. Therefore, the above statement by the authors is not true in general.**

**- "They therefore fail to predict future ice sheet behavior or feedbacks induced by external changes, such as fracture enhancement due to atmospheric or oceanic forcing"**

**At least in the case of Huth et al. (2023) this is not true. Some studies just focused on the short time scales corresponding to rifting or crevasse propagation, over which ice thickness changes are not so significant.**

Response:

Apologies for the inaccurate description. We have revised the relevant sentences in the **Introduction** section as follows (Italicized and underlined text highlights sentences that have been modified in the revised manuscript):

(Line 42–52 in the revised manuscript without tracks): "*These studies reveal the interaction between damage processes and observed ice flow dynamics. Several of them (e.g., Borstad et al., 2012; Albrecht and Levermann, 2014; Gerli et al., 2023; Sun and Gudmundsson, 2023) have one critical limitation, i.e., being diagnostic, which means that they investigate the instantaneous effect of damage on ice dynamics, but not the evolution of damage when ice thickness is allowed to evolve according to the applied changes. They therefore fail to predict future ice sheet behavior or feedbacks induced by external changes, such as fracture enhancement due to atmospheric or oceanic forcing. In contrast, more recent studies have integrated damage evolution into ice flow models to investigate fracture processes and their influence on ice dynamics, including the effects of ice thickness evolution (e.g., Duddu et al., 2020; Huth et al., 2021, 2023). These efforts primarily focus on relatively short time scales (months to years), during which ice thickness changes have limited influence on ice flow. As a result, while they shed light on instantaneous responses such as rifting or crevasse propagation, their ability to simulate long-term ice sheet feedbacks under sustained climate forcing remains limited.*"

2) **There are several notation inconsistencies as per the continuum mechanics. I am listing a few below that could identify:**

Response:

We have revised the relevant context and equations as you suggested.

**- In Eq. 1 and thereafter \tau and \epsilon are tensors so \boldsymbol should be used in LaTeX, whereas the equivalent stress is a scalar to which the n-1 is applied as exponent, so \boldsymbol should not be used here. In general vector and tensor fields are denoted using bold letters whereas scalars are denoted by**

**normal letters.**

$$2A\tau^{n-1}\boldsymbol{\tau} = \dot{\boldsymbol{\epsilon}} \quad , \tag{1}$$

**- In Eq. (2) the notation D(\tau) implies that damage D is a function of the deviatoric stress, but this fundamentally wrong. First, the definition of this 3D damage function was never even defined as eventually the depth integrated damage d based on the zero-stress theory was used. Second, the 3D damage must be defined based on the principal stress or stress invariants and not based on a component of deviatoric stress. You can refer to Pralong and Funk (2005) or Duddu et al. (2013) for more details on this. Simple fix is to say damage D without \tau in parentheses.**

$$2A\tau^{2}\boldsymbol{\tau} = (1-D)^{3}\dot{\boldsymbol{\epsilon}} \quad , \tag{2}$$

**- There are so many steps that are missing from Eq. (2) to Eq. (1) to Eq. (3), so I wonder why it is even necessary to have Eqs. (1) and (2). Perhaps, just directly start with how shallow ice flow model works and how damage is incorporated into the Kori-ULB model. Also, the parameter A is inversely related to the viscosity \mu, so I find it confusing that Eq. (3) has both A and \mu in it.**

Apologies for the confusion. Damage $D$ is incorporated into the stress balance equation through a modification to Glen's law, which described the relationship between the deviatoric stress $\tau$ and the strain rate $\dot{\epsilon}$ in the Kori-ULB model. To avoid confusion, we have removed equation (3) and revised the relevant description to clarify more logically how damage is incorporated into the Kori-ULB model.

(Line 87–118): "

*The Kori-ULB ice-sheet model (Pattyn, 2017; Coulon et al., 2024) is a vertically integrated, thermomechanical finite difference model that combines shallow-ice approximation with shallow-shelf approximation (so-called hybrid model; Winkelmann et al., 2011). The Kori-ULB ice-sheet model has been used for large-scale simulation of the AIS (Seroussi et al., 2020; Coulon et al., 2024), as well as small drainage basins with different ice geometries, such as MISMIP3d (Pattyn et al., 2013) and MISMIP+ (Cornford et al., 2020) experiments, and real-world drainage basins (e.g., Thwaites Glacier basin; Kazmierczak et al., 2024).*

*To investigate the dynamical response of the TG basin to ice damage and damage parametric perturbations, we couple the ice-sheet model with the continuum damage mechanics (CDM) model developed by Sun et al. (2017). This model establishes a direct link between the amount of damage and ice viscosity: the propagation of damage reduces the ice viscosity through Glen's flow law, leading to faster ice.*

*Following Sun et al. (2017), we represent damage using a scalar variable D, which takes values from 0 (undamaged ice) to 1 (ice entirely fractured by surface*

*and basal crevasses). The vertically averaged damage field is defined as $\overline{D}(x, y)$*

*$= d(x, y)/h \in [0,1)$, which is the closest analogue to the usual D, with $d(x, y) \in$*

*$[0,h(x, y))$ represents the vertical integral of D. Damage is incorporated into the stress balance equation through a modification to Glen's constitutive flow law. In Kori-ULB, the relationship between the deviatoric stress $\tau$ and the strain rate $\dot{\epsilon}$ is described by Glen's constitutive flow law:*

$$2A\tau^{n-1}\boldsymbol{\tau} = \dot{\boldsymbol{\epsilon}} \text{ ,} \tag{1}$$

*where A is Glen's flow law factor, dependent on the ice temperature, and n is the flow rate exponent, with n =3. And the damage feedback can be described by the integration of a damage factor D in Eq. (1):*

$$2A\tau^2\boldsymbol{\tau} = (1 - D)^3\dot{\boldsymbol{\epsilon}} \text{ ,} \tag{2}$$

*To determine the relationship between ice damage and the first principal stress, the CDM framework is based on two key components: a local source of damage term ($d_l$) that accounts for the local formation of damage, and an advection term ($d_{tr}$) that accounts for the transport of damage during ice flow.*

*In the absence of advection, ice damage is expressed as the normalized depth of the crevasses, i.e., the sum of surface crevasses $d_S$ and basal crevasses $d_b$ (Nick et al., 2011, 2013; Cook et al., 2014; Sun et al., 2017). Those can be calculated by the zero-stress assumption (Nye, 1957; Nick et al., 2011):*

$$d_s = \frac{\tau_1}{\rho_i g} + \frac{\rho_{mw}}{\rho_i} d_w, \tag{3}$$

$$d_h = \frac{\rho_i}{\rho_{sw} - \rho_i}\left(\frac{\tau_1}{\rho_i g} - H_{ab}\right), \tag{4}$$

*where $d_w$ is the water depth in the surface crevasse (here we only consider dry crevasses, so $d_w$ is equal to 0), $H_{ab}$ is the thickness above floatation, g = 9.81 m s$^{-2}$ is the gravitational acceleration, $\rho_i$ = 917 kg m$^{-3}$ is the ice density, and $\rho_{mw}$ and $\rho_{sw}$ are the densities of meltwater in the surface crevasses and seawater in the basal crevasses, respectively (both set to 1028 kg m$^{-3}$ in this study). The first principal stress $\tau_1$ is defined as the product of the first principal strain ($\dot{\epsilon}_1$) and the effective ice viscosity ($\mu$):*

$$\tau_1 = 2\mu\dot{\epsilon}_1, \tag{5}$$

*,,*

**- In Eq. (4) and (5) to be consistent the density of meltwater in the surface crevasses must be denoted differently from that of seawater in the basal crevasses. Although I understand that this study did not include meltwater in surface crevasses.**

We have revised Eqs. (4) and (5) (now Eqs. (3) and (4)) as you suggested. Please refer to our response to the previous comment as well as the revised manuscript for more details.

**- In Eq. (6) remove the star to denote multiplication, just writing C_1 h without the star implies multiplication in standard notation.**

$$d_1(\tau_1) = \min(d_\mathrm{s} + d_\mathrm{b}, C_1 h), \tag{6}$$

**- In Eq. (7) the velocity term \bold{u} is not defined. Is it a 3D or 2D velocity field? This is important as this the divergence of 3D velocity is zero but not the 2D field. Also, the terms max and min should not be italicized as they are text descriptors, and subscripts such as tr, ab, b, s should not be italicized in the equations as they are descriptors and not indices. Whereas, on Line 122, below Eq. (7), \dot{m} must be italicized as it is a variable denoting basal melting rate.**

Thank you for pointing this out. The velocity term $\boldsymbol{u}$ is the two-dimensional horizontal velocity, and we have now clarified this in the revised manuscript. In addition, we have revised Equation (7) and ensured consistency in the formatting and notation of all relevant terms throughout the manuscript, as suggested.

(Line 126–130): "

$$\frac{\partial d_\mathrm{tr}}{\partial t} + \nabla \cdot (\boldsymbol{u} d_\mathrm{tr}) = -[\max(\dot{a}, 0) + \max(\dot{m}, 0)]\frac{d_\mathrm{tr}}{h}, \tag{7}$$

*where $\boldsymbol{u}$ is the two-dimensional horizontal velocity. The left-hand side of Eq. (7) represents the conservation of vertically integrated damage, which includes the advection of crevasses with the ice flow and the effect of stretching and compression. On the right-hand side, damage reduction is modeled through two processes: an increase in undamaged ice thickness due to surface accumulation ($\dot{a}$) and erosion of the crevassed ice bottom by basal melting ($\dot{m}$).*"

3) **On line 103, it is stated that "… ice damage is expressed as the total depth of crevasses …" I suggest you say – normalized depth of crevasses – as damage is non-dimensional variable so it cannot be equated to ice thickness.**

Response:

We have revised the description as you suggested.

4) **On lines 180 and 181, the RMSE and rRMSE are denoted. I think it is better to say difference or deviation instead of error, so RMSD and rRMSD. In numerical modeling, the term error means something specific – the difference between the exact analytical solution and the approximate numerical solution.**

**Unless I misunderstood, you are reporting here are the differences between different model results.**

Response:

Thank you for your suggestion. In this study, we use error as the difference between simulations and satellite observations (though models are used to obtain those simulated values). In a strict sense it is the closest to an analytical solution we can get. In most publications in our field the term RMSE is more commonly used than RMSD. For these reasons, we have retained the term "error" in the revised manuscript.

5) **On line 206 – it is stated "For the period 1990–2020, the simulated mean net mass balance for Group 1 (with damage) is -26.5 Gt a-1, which is comparable to satellite-derived observations (-46.1 ± 7.2 Gt a-1 over 1992–2017; mean ± 1 s.d.)." It seems the satellite observations are two times larger than the simulated mean net mass balance for Group 1. Please clarify in the text here what are the reasons for this mismatch. Also, why wasn't the group 2 mass loss reported here.**

Response:

Thank you for your thoughtful comment. In this study, the Group1 and Group2 experiments are primarily distinguished based on their ability to match satellite-derived estimates of sea-level contribution (SLC) by the year 2020, rather than directly on net mass balance. We clarified this in the third paragraph of the **2.2 Simulation protocol** section, as well as summarized in Table 1 of the manuscript, where more detailed information is provided.

(Line 152–156, the third paragraph in the **2.2 Simulation protocol** section): *Based on their performance during the historical simulations, ensemble members are categorized into two subgroups according to their ability to match satellite-based estimates of ice mass change in the TG basin (Shepherd et al., 2019). Simulations where the modeled ice mass change (i.e., the contribution to sea level, SLC) falls within the satellite-derived mean estimate ± two times the observed standard deviation (s.d.) are classified as Group 1 (G1). Those that significantly over- or underestimate this range (>±2 s.d.) are classified as Group 2 (G2).*

Our results show the simulated SLC of Group 1 (with a mean of $0.24 \pm 0.04$ cm SLE) is comparable to the satellite-derived observations ($0.24 \pm 0.08$ cm). In contrast, Group 2 consists of simulations that significantly overestimate or underestimate the observed range, with the mean of $0.62 \pm 0.36$ cm SLE— approximately 2.5 times larger. As for the net mass balance, although there remains a discrepancy between the satellite observations and the simulation results of Group 1, the mean net mass balance in Group 1 ($-26.5$ Gt $a^{-1}$) is considerably closer to satellite-derived estimates ($-46.1 \pm 7.2$ Gt $a^{-1}$) than simulations that neglect the

damage process (1.2 Gt a$^{-1}$), or those in Group 2, which can reach $-169$ Gt a$^{-1}$ under extreme damage scenarios. Given the inherent uncertainties and simplifications in ice-sheet modeling, it is challenging for simulations to exactly reproduce observational data. Therefore, in this study, we focus primarily on the performance of our historical simulations that can capture the sea-level contribution rather than the net mass balance.

We have not discussed the result of Group 2 in detail in the main text. The simulated SLCs from Group 2 are outside the range of observational estimates $\pm$ 2 s.d. in the historical simulation. Therefore, we consider the parameters used to represent the damage process in Group 2 experiments to be unrealistic. Nonetheless, they are included in all relevant figures for comparison, allowing us to evaluate differences between the two groups.

6) **If Fig8. even in the extreme experiments the SLC is less than 18 cm by 2200. Thwaites is referred to as the doomsday glacier in some news articles, perhaps a comment can be added on this result in the context of how catastrophic the projected SLC of 18 − 24 cm is.**

Response:

Thank you for your comment. In our simulations, we use present-day atmospheric and oceanic forcing in our forward simulations rather than projections based on future climate scenarios. Our results show that, even under extreme damage scenarios, sea-level contribution (SLC) increases from 1–2 cm (without damage) to approximately 18 cm by 2200. Therefore, by considering the combined effect of damage and future climate change, Thwaites Glacier is likely to reach MISI if the grounding line retreats too much. It is not a matter of sea-level content, but rather, sea-level rise acceleration, and we do observe acceleration.

We have added some sentences to discuss this limitation of our study in the **Discussion** section of the revised manuscript as follows.

(Line 446–451): "*In our forward simulations, present-day atmospheric and oceanic forcing are applied, rather than projections based on future climate scenarios. Our results show that, under extreme damage scenarios, the sea-level contribution (SLC) from Thwaites Glacier increases from 1 cm (without damage) to approximately 18 cm by 2200 (Fig. 8a), indicating that damage alone can substantially accelerate mass loss leading to collapse. This suggests that the combined effects of ice damage and future climate changes could further enhance mass loss from Thwaites Glacier, reinforcing its potential role as a major contributor to future sea-level rise.*"

7) **On line 360, it is stated that "As damage is advected with the ice flow, this**

**fraction increases toward the ice front, reaching 0.3 in lower-damage cases to 0.7 in higher-damage cases, with particularly high damage concentrated in the shear zone." My intuition was that the increasing strain rate toward the ice front is the major contributor to this damage increase and not the advection. Please clarify the relative contribution of advection and damage nucleation to the increase in damage downstream.**

Response:

Apologies for the confusion. We agree that both advection and strain-rate-dependent damage nucleation contribute to the downstream increase in damage. In our model, damage evolves through two primary mechanisms: (1) local damage nucleation and growth, which is strongly controlled by the strain-rate regime and (2) advection of existing damage with the ice flow. While our original wording emphasized advection, we acknowledge that strain-rate-driven damage nucleation could also play a dominant role, especially near the ice front where tensile and shear strain rates are elevated—particularly within shear margins. However, due to the strong interaction between the advection and damage nucleation in our model, we currently lack an effective method to explicitly decipher their relative contributions to the increase in downstream damage in this study. For instance, damage advection not only directly affects the final damage field, but also indirectly impacts it by affecting the local strain rate (and ice velocity), and the subsequently local damage nucleation. In other words, the local damage nucleation due to high strain rate has already been affected by damage advection. Conversely, local strain rate can also affect damage advection. Therefore, it is not easy to quantify the relative contributions of local damage nucleation driven by increased strain rates and damage advection to the final damage field.

To avoid confusion, we have revised the corresponding sentence to more accurately reflect the combined influence of these two processes:

(Line 372–374): "*As ice flows toward the front, damage increases due to the combined effects of advection and local damage nucleation driven by increased strain rates. Damage fractions range from 0.3 in lower-damage cases to 0.7 in higher-damage cases, with particularly high damage concentrated in the shear zone.*"

8) **On line 409, it is stated "Instead of solely relying on ice sheet mass loss data, future efforts should incorporate observational datasets of crevasse distributions." I do not disagree with this suggestion but there is a more nuanced discussion missing here. From our modeling studies, we find that most of the damage in ice shelves must be in the form of basal crevasses, especially if there is no hydrofracture in surface crevasses. Due to the ocean water pressure at the terminus, there is simply not enough driving force in floating ice shelves to propagate surface cracks deeper into the ice below the waterline.**

**Our studies on Larsen C ice shelf in Huth et al. (2023) indicate that rift propagation is almost entire driven by basal crevasses formation. Also, Clayton et al. (2024) shows that crevasses can propagate deeper in ice shelves due to the less dense firn layers near the top surface. There are no observations of basal crevasses, and the extent of firn layer is not so well quantified that can rightly inform future modeling efforts. My comment is that this requires basin/ice-shelf scale process modeling (e.g. full Stokes and phase field fracture) to better understand and represent basal crevasses evolution.**

**Clayton, T., Duddu, R., Hageman, T., & Martínez-Pañeda, E. (2024). The influence of firn layer material properties on surface crevasse propagation in glaciers and ice shelves. The Cryosphere, 18(12), 5573-5593.**

Response:

Thank you for this constructive comment. We agreed with your important insights regarding the critical roles of basal crevasse formation and the firn layers near the top surface in driving rift and crevasses propagation within ice shelves. We have revised the relevant sentences to include this in the discussion here.

(Line 423–429): "*Instead of solely relying on ice sheet mass loss data, future efforts should incorporate observational datasets of crevasse distributions. A critical limitation remains the lack of direct observations of basal crevasses and the uncertainty in quantifying firn layer structure, both of which hinder accurate representation of damage processes. Recent studies suggest that basal crevasses may play a dominant role in damage evolution, particularly in the absence of surface hydrofracture (Huth et al., 2023), while firn properties can also influence crevasse penetration depth in ice shelves (Clayton et al., 2024). Therefore, future efforts should also prioritize high-resolution, basin- or ice-shelf-scale process modeling—such as phase-field fracture models—to better understand and represent the evolution of basal crevasses.*"

9) **On lines 412, it is stated "hindcasts for 1990–2020 (Schimdtko et al., 2014; Kittel et al., 2021) do not necessarily reflect the actual imbalance of the ice sheet during that period." I did not understand this sentence. Please explain this in more detail. What does it mean to do hindcast simulations and which figures in the paper show these results.**

Response:

Sorry for the confusion. We have deleted this sentence.

10) **On line 424, it is stated that "Our results suggest that ice damage could be a key driver of Thwaites Glacier's rapid ice loss, offering an alternative explanation to previous hypotheses." Please clarify what is meant by rapid ice**

**loss. The sea level contribution is 18 - 24 cm by 2300 in the extreme scenarios, which is significant, but does it warrant the use of the term "rapid."**

Response:

Thank you for pointing this out. To avoid confusion, we have deleted the term "rapid".

**11) Line 440, instead of saying "increasing damage intensity …" it is clearer to say "an increase in damage intensity …"**

Response:

We have revised the sentence as you suggested.

**12) Overall, the figures are quite well made, and the caption are comprehensive and informative. However, I have a few minor questions or suggestions below.**

**- I do not understand what the length white-gray, black-gray bars in subfigures 1(a), 1(b) and 1(d). Also, what the black regions in (a) and (b), are these the ocean regions or grounded regions that are removed from the images.**

Response:

Thank you for your comment. The white-gray, black-gray bars in subfigures 1(a), 1(b), and 1(d) are linear scale bars, depicted as marked line segments with numerical labels that visually convey the map's scale. For example, in Fig. 1a, the bar is labeled 0, 50 km, and 100 km, indicating that each segment represents 50 km of actual distance. We have added a sentence in the caption of Fig.1 to explain the white-gray and black-gray bars.

(Line 207): "*The black-gray bar in panel (a) and white-gray bars in panels (c) and (d) are scale bars of the corresponding maps.*"

The black regions in (a) and (b) are the ocean regions shown in the Landsat Image Mosaic of Antarctica (LIMA; Bindschadler et al., 2008).

**- In Fig. 2a, I do not see the black line corresponding to observational estimates.**

The observational estimate and its ±1 standard deviation range are represented by vertical black lines on the right side of Fig. 2a. We have revised the figure caption to explicitly state this.

(Line 239–241): "*The vertical black lines on the right side of panel (a) represent the observed mean value ± 1 standard deviation (Shepherd et al., 2019). The black line and shaded area in panel (b) represent the observed mean values ± 1 standard deviation (Shepherd et al., 2019).*"

**- In Fig. 10 there are positive ice thickness changes (red regions in second and third figure columns) ahead of the ice shelves. Is that due to ice mélange changes or is that something that is unphysical. Please clarify in the caption.**

Thank you for your comment. The positive thickness change ahead of the ice shelves is due to an advance of the calving front. Typically, in higher-damage simulations, there is a strong increase in velocity and hence a following advance of the calving front. To clarify this, we have added the following sentence to the caption of Fig. 10:

(Line 387–389): "*The positive thickness change observed ahead of the ice shelves is caused by the advance of the calving front, which results from increased ice velocity under higher-damage scenarios.*"

In the Kori-ULB ice-sheet model, calving at the ice front depends on the combined penetration depths of surface and basal crevasses relative to total ice thickness. The depths of the surface and basal crevasses are parameterized as functions of the divergence of ice velocity, the accumulated strain, the ice thickness, and (optionally) surface liquid water availability, similar to Pollard et al. (2015) and DeConto and Pollard (2016). In this study, we used this calving parameterization, but without considering hydrofracturing. A description of how calving is implemented in our model has been added to the revised manuscript for clarity:

(Line 165–168): "*Calving at the ice front is determined by the combined penetration depths of surface and basal crevasses relative to ice thickness, with crevasse depths parameterized as functions of ice velocity divergence, accumulated strain, ice thickness, and (optionally) surface liquid water availability, similar to Pollard et al. (2015) and DeConto and Pollard (2016).*"

Pollard, D., DeConto, R. M., and Alley, R. B.: Potential Antarctic Ice Sheet retreat driven by hydrofracturing and ice cliff failure, Earth and Planetary Science Letters, 412, 112–121, doi:10.1016/j.epsl.2014.12.035, 2015.

DeConto. R. M. and Pollard, D.: Contribution of Antarctica to past and future sea-level rise, Nature, 531, 591-597, doi:10.1038/nature17145, 2016.